# Charcot–Marie–Tooth type 2A variants of mitofusin 2 sensitize cells to apoptotic cell death

Mariana Joaquim[1,2,3,*], Maria-Bianca Bulimaga[1,2,3,4,*], Marie A. Mohn[1,2], Solenn Plouzennec[5], Leon Osinski[1,2], Selver Altin[1,2], Esther Mahabir[6], Arnaud Chevrollier[5] and Mafalda Escobar-Henriques[1,2,3,‡]

## ABSTRACT

The neuropathy Charcot–Marie–Tooth (CMT) is an incurable disease with a lack of genotype–phenotype correlation. Variants of the mitochondrial protein mitofusin 2 (MFN2), a large GTPase that mediates mitochondrial fusion, are responsible for the subtype CMT type 2A (CMT2A). Interestingly, beyond membrane remodelling, additional roles of MFN2 have been identified, expanding the possibilities to explore its involvement in disease. Here, we investigated how cellular functions of MFN2 are associated with variants present in individuals with CMT2A. Using human cellular models, we observed that cells expressing CMT2A variants display increased endoplasmic reticulum (ER) stress and apoptotic cell death. Increased cleavage of PARP1, caspase 9, caspase 7 and caspase 3, alongside BAX translocation to mitochondria, pointed towards effects on intrinsic apoptosis. Moreover, although disruption of fusion and fission dynamics per se did not correlate with cell death markers, expression of MFN1 or MFN2 alleviated the apoptosis markers of CMT2A variant cell lines. In sum, our results highlight excessive cell death by intrinsic apoptosis as a potential target in CMT2A disease.

KEY WORDS: Charcot–Marie–Tooth, CMT2A, MFN2, Apoptosis, Cell death, Mitochondria, Fusion

## INTRODUCTION

Charcot–Marie–Tooth (CMT), the most commonly inherited neuropathy, is a peripheral nervous system disorder characterized by chronic motor and sensory polyneuropathy (Bird, 1993; Bolino and D'Antonio, 2023) that, as yet, has no cure. It comprises a group of phenotypically and genetically heterogenous diseases, with more than 100 causative genes identified (Pipis et al., 2019). Interestingly,

despite the vast genetic heterogeneity, the clinical manifestations and respective symptomatic treatments are remarkably similar. Axonal forms of CMT fall into the CMT2 category, with variants of the mitochondrial protein mitofusin 2 (MFN2) being the most common underlying cause (Zuchner et al., 2005). Individuals with CMT2 that carry pathogenic variants in the *MFN2* gene are categorized as having CMT type 2A (CMT2A), which accounts for ∼90% of the most severely impaired individuals with CMT2 (Feely et al., 2011). To date, more than 300 variants in hundreds of residues have been identified, which distribute throughout MFN2 and do not appear to affect specific protein domains (Fig. 1A,B) (Stojkovic, 2016). CMT2A can be inherited in an autosomal dominant (∼90%) (OMIM: #609260) or in a recessive (∼10%) manner (OMIM: #617087) (Zuchner et al., 2005). It mostly presents at an early onset age and is characterized by progressive muscle weakness and atrophy and by sensory loss (Zuchner et al., 2005). Additionally, despite being a peripheral neuropathy, the most severe cases of CMT2A are described to be accompanied by central nervous system deficits. To date there is a considerable knowledge gap due to the lack of correlation between genotype and phenotype. Strikingly, variants within the same domain, or even in the same residue, can lead to different degrees of disease severity (Beręsewicz et al., 2018). As for all CMT disorders, CMT2A is incurable, and only symptomatic-based therapeutic approaches are available to date.

The mitofusins MFN1 and MFN2 are large dynamin-like GTPase proteins (DRPs) that allow fusion of outer mitochondrial membranes by self-oligomerization. Their GTPase domains are located at the N terminus, followed by one hydrophobic heptad repeat (HR1), the transmembrane anchor(s) and finally a second protein–protein interaction domain, HR2 (Daumke and Praefcke, 2016) (Fig. 1A). Along with mitofusins, other DRPs mediate mitochondrial dynamics: optic atrophy 1 (OPA1) controls fusion of inner mitochondrial membranes, whereas dynamin-related protein 1 (DRP1, also known as DNML1), a cytosolic family member that transiently localizes to the mitochondrial surface, controls mitochondrial fission (Quintana-Cabrera and Scorrano, 2023). Intriguingly, despite the clear and major role of MFN2 in mediating mitochondrial fusion, membrane morphology defects do not seem to be the underlying cause of CMT2A. Although several *MFN2* variants are associated with a fragmented mitochondrial network (Detmer and Chan, 2007; Franco et al., 2020; Misko et al., 2010; Wolf et al., 2019) or even result in mitochondrial hyperfusion (Das et al., 2022; Samanas et al., 2020), several others do not affect mitochondrial morphology (Amiott et al., 2008; Detmer and Chan, 2007; Larrea et al., 2019). Additionally to mitochondrial membrane remodelling, MFN2 has also been implicated in mitochondrial respiration, motility, contact to the endoplasmic reticulum (ER), mitophagy and apoptosis (Bernard-Marissal et al., 2019; Larrea et al., 2019; Rizzo et al., 2016; Saporta et al., 2015; Zaman and Shutt, 2022). Moreover,

[1]Institute for Genetics, University of Cologne, 50674 Cologne, Germany. [2]Cologne Excellence Cluster on Cellular Stress Responses in Aging-Associated Diseases (CECAD), University of Cologne, 50931 Cologne, Germany. [3]Center for Molecular Medicine Cologne (CMMC), University of Cologne, 50931 Cologne, Germany. [4]Institute of Pathology, Medical Faculty and University Hospital, University of Cologne, 50937 Cologne, Germany. [5]University of Angers, MitoLab Team, MitoVasc Unit, CNRS UMR6015, INSERM U1083, Structure Fédérative de Recherche (SFR), Interactions Cellulaires et Applications Thérapeutiques (ICAT), 49330, Angers, France. [6]Comparative Medicine, Center for Molecular Medicine Cologne (CMMC), Medical Faculty and University Hospital, University of Cologne, 50931 Cologne, Germany.
*These authors contributed equally to this work

‡Author for correspondence (Mafalda.Escobar@uni-koeln.de)

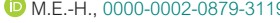 M.E.-H., 0000-0002-0879-3119

Journal of Cell Science

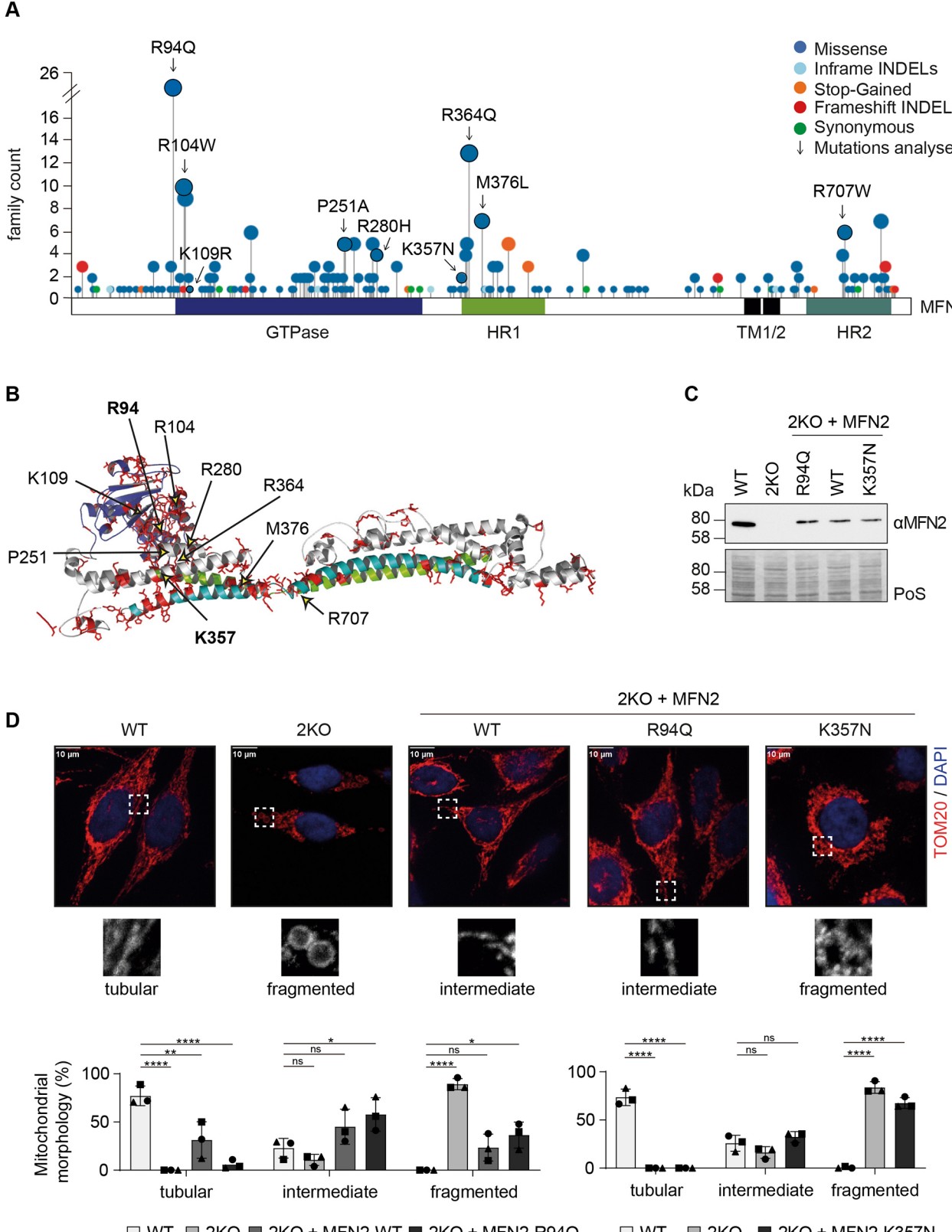

**Fig. 1.** See next page for legend.

distinct CMT2A variants are reported to impact these cellular processes in a different manner (Zaman and Shutt, 2022). Thus, assigning MFN2 with a specific disease gatekeeper role does not seem to be straightforward.

Here, we investigated the molecular basis for MFN2 involvement in CMT2A, taking advantage of the growing knowledge on its various roles in the cell (Escobar-Henriques and Joaquim, 2019; Joaquim and Escobar-Henriques, 2020). Our findings are consistent

**Fig. 1. CMT2A variants distribute throughout the MFN2 protein and cause different mitochondrial morphologies.** (A) Locations and frequency of occurrence of CMT2A variants in a linear representation of the MFN2 protein. The domains of the protein are labelled, and the variants analysed in this study are indicated with arrows. GTPase (blue), GTPase domain; HR1 (light green) and HR2 (dark green), heptad repeat domains; TM 1/2 (black), transmembrane domains. INDEL, insertion–deletion variant. (B) Structural localization of CMT variants in full length MFN2, modelled on the structure of GMPPNP-bound BLDL (PDB ID: 2W6D; Low et al., 2009), as previously reported (Anton et al., 2019). The positions of variants analysed in this study are identified with arrows. (C) Western blot analysis of HeLa WT cells, 2KO cells, and 2KO cells stably expressing FLAG-tagged WT MFN2 or MFN2 variants (R94Q or K357N), immunoblotted with anti-MFN2. Staining of total protein with Ponceau S (PoS) was used as loading control. Blots shown are representative of three experiments. (D) Confocal images, after immunostaining with the outer mitochondrial membrane protein TOM20 (in red) and DAPI (in blue), of HeLa WT cells, 2KO cells, and 2KO cells stably expressing WT MFN2 or MFN2 variants (R94Q or K357N). Dashed white boxes indicate regions shown as expanded views below the main images, which show examples of the most common morphological category for each cell line. Mitochondrial morphology was categorized as tubular, intermediate or fragmented. The number of cells showing the respective mitochondrial phenotypes is shown in the graphs, expressed as a percentage of the total number of cells. The experiments containing the two variants were performed independently and hence are presented in two separate graphs. At least 50 cells of three biological replicates were counted for quantification of mitochondrial morphology. Individual values of each experiment are plotted as black filled triangles, circles or squares. The bars represent the mean percentage of cells with each mitochondrial morphology±s.d. ($n$=3 biological replicates). Scale bars: 10 μm. Two-way ANOVA with Tukey's post-hoc tests was applied. $P$-values of left graph from left to right: ****$P$<0.0001; **$P$=0.0036; ****$P$<0.0001; ns (not significant), $P$=0.9765; ns, $P$=0.4953; *$P$=0.0472; ****$P$<0.0001; ns, $P$=0.4212; *$P$=0.0343. $P$ values of right graph from left to right: ****$P$<0.0001; ****$P$<0.0001; ns, $P$=0.4966; ns, $P$=0.8675; ****$P$<0.0001; ****$P$<0.0001.

with the reported lack of correlation between CMT2A variants and mitochondrial morphology alterations (Zaman and Shutt, 2022). Instead, we found that CMT2A variants cause an increased propensity for intrinsic apoptosis. In contrast, absence of MFN1, MFN2 or of the main fission factor DRP1 did not correlate with cell death. Finally, although CMT2A variants present increased ER stress, inhibition of the integrated stress response (ISR) did not significantly alter their cell death propensity. In contrast, boosting either MFN1 or MFN2 expression levels alleviated cell death of our CMT2A human cell model.

## RESULTS
### Mitochondrial morphology does not correlate with CMT2A
To systematically assess how CMT2A variants generally affect MFN2 functions, we profited from our recently characterized MFN2 knockout model in HeLa cells (2KO; Joaquim et al., 2025). We started by analysing the correlation between CMT2A and mitochondrial morphology. We evaluated the effect on mitochondrial morphology of nine MFN2 variants with different locations in MFN2 or frequency in individuals with CMT2A (Fig. 1A,B). Immunocytochemistry analysis of HeLa cells, transiently expressing FLAG-tagged versions of the selected MFN2 variants, showed that CMT2A variants led to a range of outcomes in mitochondrial morphology (Fig. S1). This confirmed previous reports showing that altered mitochondrial dynamics is not a trait common to all MFN2 disease variants (Zaman and Shutt, 2022). For example, transient expression of the MFN2 variant R94Q partially rescued mitochondrial tubulation almost as well as wild-type (WT) MFN2, whereas expression of K357N MFN2 failed to do so (Fig. S1). To independently evaluate the effects of MFN2

variants on mitochondrial morphology, while avoiding possible artefacts attributable to transient transfection analyses, we created HeLa cell lines stably expressing these two CMT2A variants. Stable expression of WT MFN2 or the disease variants R94Q or K357N was achieved by infection of HeLa 2KO cells with lentiviruses carrying FLAG-tagged constructs (MFN2[FLAG], MFN2[R94Q-FLAG] and MFN2[K357N-FLAG], respectively) followed by their monoclonal selection. Stable cell line clones presenting similar levels of the MFN2 protein were chosen (Fig. 1C). Mitochondrial morphology of the stable cell lines was analysed by immunocytochemistry with the outer mitochondrial membrane protein TOM20 (also known as TOMM20). As expected, WT HeLa cells displayed a tubular mitochondrial network, which appeared severely fragmented in the absence of MFN2 (Fig. 1D; Fig. S2). Stable re-expression of WT MFN2 partially rescued mitochondrial tubulation (Fig. 1D), with the cells resembling WT cells with regard to mitochondrial junctions and mitochondrial length (Fig. S2A). Similar analysis revealed that 2KO cells expressing the R94Q and K357N variants had defective mitochondrial morphology, with cells expressing K357N being more similar to 2KO cells (Fig. 1D; Fig. S2). In sum, different CMT2A variants in MFN2 result in distinct mitochondrial morphology patterns. Therefore, changes in mitochondrial morphology do not appear to be the underlying cause of CMT2A.

### CMT2A variants of MFN2 cause cell death
Beyond mitochondrial fusion, MFN2 has also been shown to have an impact on quality control processes, being, for example, linked to apoptosis (Brooks et al., 2007; Han et al., 2020; Hoppins et al., 2011; Jin et al., 2011; Joaquim and Escobar-Henriques, 2020; Neuspiel et al., 2005; Wei et al., 2020). We therefore assessed the MFN2[R94Q-FLAG] and MFN2[K357N-FLAG] stable cell lines for the presence of cell death markers. We started by analysing the presence of cleaved poly [ADP-ribose] polymerase 1 (PARP1) protein, a well-known apoptotic marker (Zhu et al., 2023). PARP1 was found to be constitutively cleaved in cells stably expressing either the R94Q or the K357N MFN2 variant but not WT MFN2 (Fig. 2A). Interestingly, PARP1 cleavage in MFN2[K357N-FLAG] cells was alleviated by overexpression of either WT MFN1 or MFN2 (Fig. 2B). This is in agreement with previous observations showing amelioration of CMT2A-like symptoms upon boosting of MFN1 levels, both in cultured neurons and in a mouse model (Misko et al., 2012; Zhou et al., 2019), possibly occurring via mitofusin oligomerization (Detmer and Chan, 2007). To strengthen our findings, we investigated whether CMT2A variants caused increased cleaved PARP1 in another cell line. Indeed, even though PARP1 cleavage is technically more challenging to detect in HEK293 cells due to the low basal levels of uncleaved PARP1, stable expression of the K357N variant also increased PARP1 cleavage in HEK293, reproducing the phenotype of HeLa cells (Fig. 2C). In sum, CMT2A variants of MFN2 increase cell death.

### Disruption of mitochondrial fusion and fission does not drive death of HeLa cell models
Although the CMT2A variant MFN2 K357N caused a significant increase in PARP1 cleavage, increased cleavage of PARP1 was not observed in 2KO cells (Fig. 2A). However, mitochondrial morphology was similarly affected in both cases (Fig. 1D; Fig. S2), which suggested that disruption of mitochondrial morphology per se did not induce cell death. To independently analyse the effects of altering mitochondrial shape on cell death, we tested PARP1 cleavage in the absence of MFN1 or upon knockdown or knockout of DRP1. Indeed, preventing

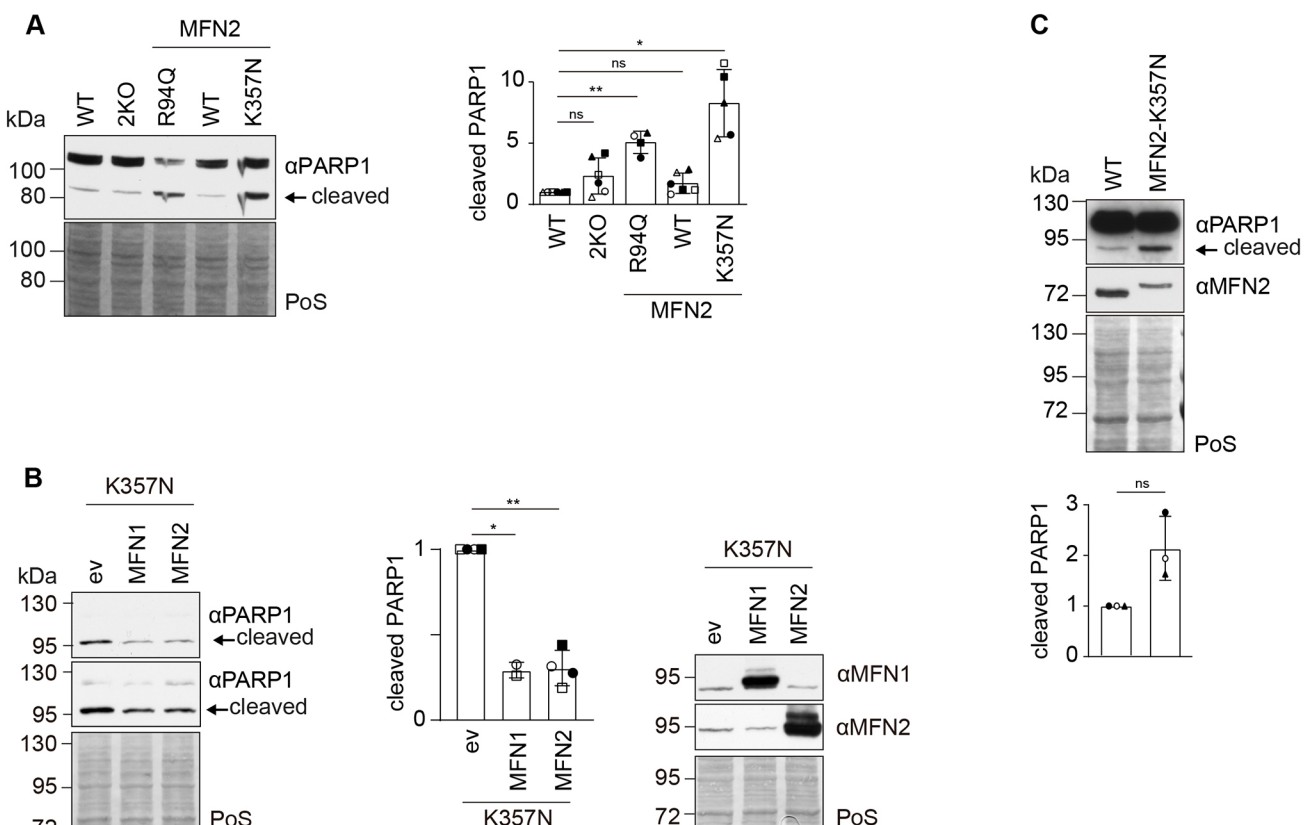

**Fig. 2. Cells expressing CMT2A variants of MFN2 have increased cleavage of PARP1, a cell death marker.** (A) Western blot analysis (left panel) and quantification (right panel) of HeLa WT cells, 2KO cells, and 2KO cells stably expressing WT MFN2 or MFN2 variants (R94Q or K357N), immunoblotted with anti-PARP1. Staining of total protein with Ponceau S (PoS) was used as loading control. Bars represent the mean±s.d. intensity of cleaved PARP1, normalized to the PoS staining and shown relative to WT ($n$=4–6 biological replicates). Individual values of each experiment are plotted as filled and open triangles, circles or squares. Mixed effect statistical analysis was applied. $P$ values from left to right: ns (not significant), $P$=0.3093; **$P$=0.0090; ns, $P$=0.3285; *$P$=0.0161. (B) Western blot analysis (left and right panels) and quantification (middle panel) of HeLa 2KO cells stably expressing the K357N variant of MFN2, transiently transfected with empty vector (ev), MFN1 or MFN2, and immunoblotted with anti-PARP1 (left), anti-MFN1 or anti-MFN2 (right). It should be noted that submitting the cells to the transient transfection mix already increased PARP1 cleavage. Staining of total protein with PoS was used as a loading control. Bars represent the mean±s.d. intensity of cleaved PARP1, normalized to the PoS staining and shown relative to the ev control ($n$=2–4 biological replicates). Individual values of each experiment are plotted as open and filled circles and squares. Mixed effect statistical analysis was applied. $P$ values from left to right: *$P$=0.0328, **$P$=0.0019. (C) Western blot analysis (upper panel) and quantification (lower panel) of control HEK 2KO cells (WT) and HEK 2KO cells stably expressing the K357N variant of MFN2, immunoblotted with anti-PARP1 and anti-MFN2 antibodies. Staining of total protein with PoS was used as loading control. Bars represent the mean±s.d. intensity of cleaved PARP1, normalized to the PoS staining and shown relative to WT ($n$=3 biological replicates). Individual values of each experiment are plotted as open and filled circles and triangles. Two-tailed paired $t$-test was applied. ns, $P$=0.0891.

mitochondrial fusion or fission did not alter the propensity for PARP1 cleavage (Fig. 3A; Fig. S3A). To confirm the role of CMT2A variants in cell death, we analysed PARP1 cleavage in two cell lines, originating from single-cell selection, stably expressing the MFN2 K357N variant at different protein levels. Once again, both MFN2[K357N-FLAG] clones exhibited PARP1 cleavage, even in the clone where MFN2 K357N was present at a lower level than endogenous WT MFN2 in WT HeLa cells (Fig. S3B). These results reinforced a novel property of CMT2A variants dissociated from the levels of MFN2 or alterations in mitochondrial shape. As previously observed in other cell lines (Das et al., 2022; Joaquim et al., 2025; Murata et al., 2020), the levels of mitochondrial DRPs are interdependent (Fig. S3A). Therefore, we tested whether CMT2A variants also regulate levels of fusion and fission components. However, no significant changes could be found in either the fusion factors MFN1 and OPA1, or the fission factors DRP1 and MFF (Fig. 3B,C). Interestingly, cells treated with actinomycin D (ActD), a transcriptional inhibitor that induces the apoptotic machinery (Liu

et al., 2016; Lu et al., 2015), presented significant alterations in the band pattern of the fusion factor OPA1, with a decrease in the long forms and an increase in the short forms (Fig. S3C). This is a well-studied response to several stress situations that cause mitochondrial fragmentation (Anand et al., 2014). We therefore checked whether the alleviation of cell death observed upon overexpression of MFN1 or MFN2 in MFN2[K357N-FLAG] cells (Fig. 2B) also impinged on the OPA1 protein cleavage profile. However, in this case no major alterations could be observed (Fig. S3D). In sum, although CMT2A variants of MFN2 acquire a pro-cell death propensity, this does not occur in cells lacking fusion or fission components.

## Cell death in CMT2A variant cell lines is caspase dependent

We then aimed at analysing other evidence of cell death in our CMT2A cell model. First, qualitative analysis of morphological alterations, by transmission electron microscopy, revealed the presence of some cells with typical cell death signs reminiscent of apoptosis in the MFN2[K357N-FLAG] cell line (Fig. S4A). Therefore,

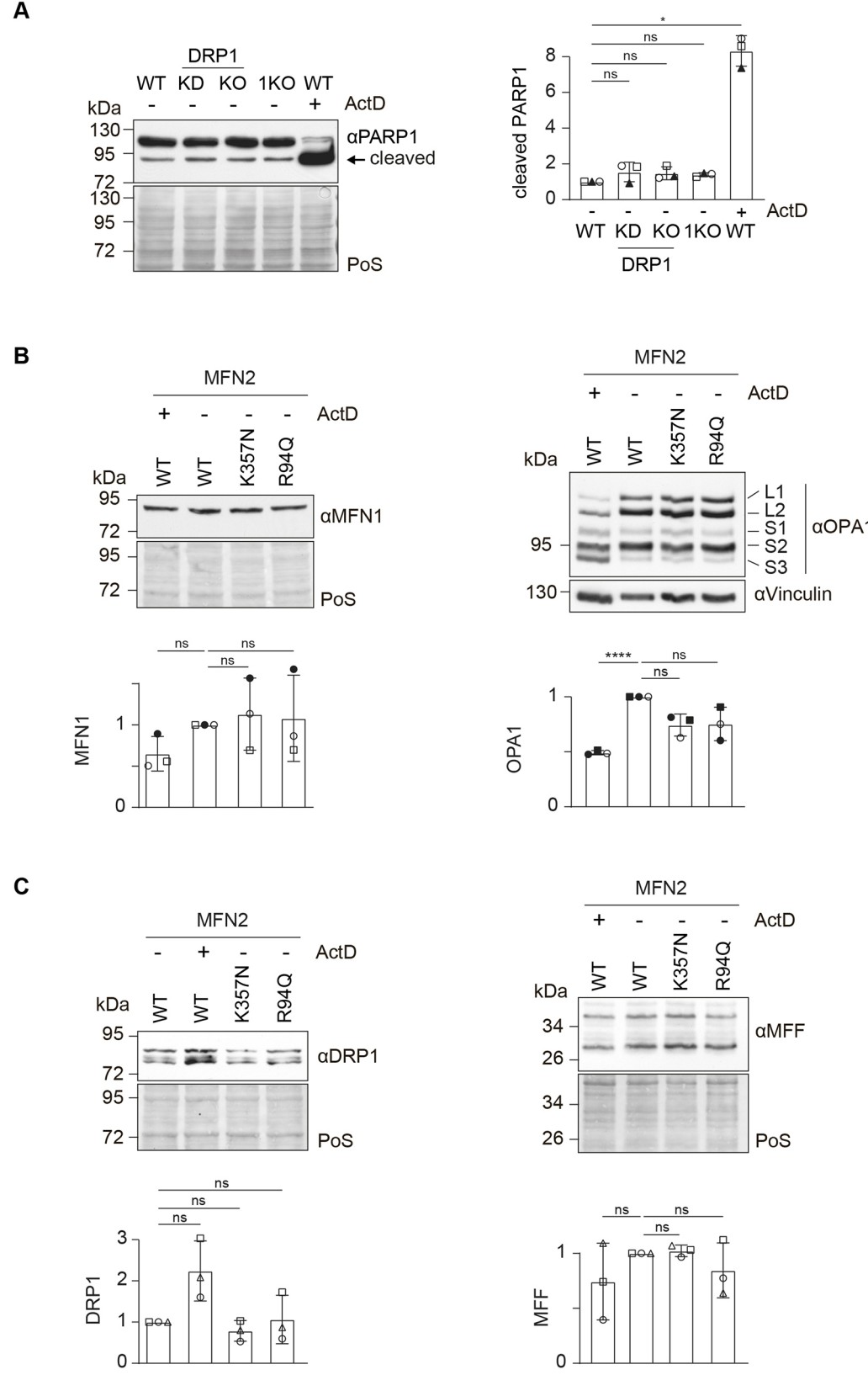

**Fig. 3.** See next page for legend.

we investigated whether increased PARP1 cleavage in our CMT2A cell model was a result of apoptosis, by inhibiting this process with a pan-caspase inhibitor, Z-VAD-FMK (ZVAD). This is because

besides being subject to caspase cleavage (Zhu et al., 2023), PARP1 is also targeted by non-caspase proteases including calpains, cathepsins, granzymes and matrix metalloproteases (Chaitanya

**Fig. 3. Mitochondrial dynamics proteins are unaffected by expression of CMT2A variants of MFN2.** (A) Western blot analysis (left panel) and quantification (right panel) of HeLa WT cells either untreated or treated with ActD (1 µM, 6 h), DRP1 knockdown cells (KD), DRP1 knockout cells (KO) and MFN1 KO cells (1KO), immunoblotted with anti-PARP1 antibody. Staining of total protein with Ponceau S (PoS) was used as loading control. Bars represent the mean±s.d. intensity of cleaved PARP1, normalized to the PoS staining and shown relative to the untreated WT group (n=3 biological replicates). Individual values of each experiment are plotted as open and filled circles, squares and triangles. One-way ANOVA with Tukey's post-hoc tests was applied. P values from left to right: ns (not significant) P=0.5577; ns, P=0.4067; ns, P=0.0652; *P=0.0140. (B) Western blot analysis (upper panels) and quantification (lower panels) of HeLa 2KO cells stably expressing FLAG-tagged WT MFN2, either untreated or treated with ActD (1 µM, 6 h), and 2KO cells stably expressing FLAG-tagged variants of MFN2 (R94Q or K357N), immunoblotted with anti-MFN1 (left) and anti-OPA1 (right) antibodies. Staining of total protein with PoS was used as loading control for MFN1. Immunoblot with anti-vinculin antibody was used as loading control for OPA1. Bars represent the mean±s.d. intensity, normalized to the PoS staining and shown relative to untreated HeLa 2KO cells stably expressing FLAG-tagged WT MFN2 (n=3 biological replicates). Individual values of each experiment are plotted as circles and squares. One-way ANOVA with Tukey's post-hoc tests was applied. P values for MFN1 quantification from left to right: ns, P=0.2404; ns, P=0.9916; ns, P=0.9466. P values for OPA1 quantification from left to right: ***P<0.0001; ns, P=0.2488; ns, P=0.1164. (C) Western blot analysis (upper panels) and quantification (lower panels) of HeLa 2KO cells stably expressing FLAG-tagged WT MFN2, either untreated or treated with ActD (1 µM, 6 h), and 2KO cells stably expressing FLAG-tagged variants of MFN2 (R94Q or K357N), immunoblotted with anti-DRP1 (left) and anti-MFF (right) antibodies. Staining of total protein with PoS was used as loading control. Bars represent the mean±s.d. intensity, normalized to the PoS staining and shown relative to untreated HeLa 2KO cells stably expressing FLAG-tagged WT MFN2 (n=3 biological replicates). Individual values of each experiment are plotted as open circles, squares and triangles. One-way ANOVA with Tukey's post-hoc tests was applied. P values for DRP1 quantification from left to right: ns, P=0.2312; ns, P=0.9969; ns, P=0.5744. P values for MFF quantification from left to right: ns, P=0.6521; ns, P=0.7385; ns, P=0.8684.

et al., 2010). As a positive control, WT cells were subjected to ActD treatment, both in presence and absence of ZVAD. As expected, ZVAD prevented PARP1 cleavage caused by ActD (Fig. 4A). Similarly, caspase inhibition with ZVAD led to a rescue of PARP1 cleavage in MFN2$^{K357N-FLAG}$ cells (Fig. 4A), pointing to a basal increase of apoptotic cell death in CMT2A cell models.

To be able to quantitatively determine cell death rates, we employed the Incucyte live-cell analysis system. Cells were imaged for 48 h by staining with Annexin V 568 dye, a fluorescently labelled molecule that binds to phosphatidylserine upon apoptosis-induced cellular permeabilization. Cells were analysed either in the absence or presence of apoptotic inducers consisting of a combination of the BH3 mimetic ABT-737 and the MCL1 inhibitor S63845 (referred to from now on as ABT+S). Addition of 1 µM ABT+S induced cell death both in WT and in MFN2$^{K357N-FLAG}$ cells as expected (Fig. S4B). Reduction of the ABT+S concentration to 0.1 µM induced cell death in MFN2$^{K357N-FLAG}$ cells. However, although sufficient to cause cell death in MFN2$^{K357N-FLAG}$ cells, this low dosage treatment did not affect WT cells (Fig. 4B), underlining an increased propensity of MFN2$^{K357N-FLAG}$ cells to undergo cell death. Moreover, simultaneously adding ZVAD and 0.1 µM ABT+S prevented the death of MFN2$^{K357N-FLAG}$ cells, confirming a caspase-dependent effect (Fig. 4B). In conclusion, cells with CMT2A variants are more sensitive to apoptosis. Additionally, we evaluated cell death upon expression of other MFN2 disease variants, R364Q and R707W, in HeLa 2KO cells. Indeed, transient expression of all CMT2A variants led to increased cell death when compared to expression of WT MFN2 (Fig. 4C).

Together, these results show that CMT2A variants increase cell death by apoptosis.

## CMT2A variants have increased ER stress, which is not linked to their role in cell death

Similar to the phenotypes observed for cells expressing CMT2A variants, ER stress has been shown to promote PARP1 cleavage and lead to extrinsic apoptotic cell death (Lindner et al., 2020; Rao et al., 2002). Consistent with this, addition of the ER stressor thapsigargin (Tg), which blocks sarco/ER Ca$^{2+}$-ATPase (SERCA) pumps (Oslowski and Urano, 2011), caused a clear induction of PARP1 cleavage (Fig. 5A). Hence, we investigated whether ER stress was increased in CMT2A variant-expressing cells. To this aim, the levels of well-known ER stress markers were measured by reverse transcription quantitative PCR (qPCR) analysis. Indeed, *CHOP1* (also known as *DDIT3*), *ATF4*, spliced *XBP1* (*XBP1s*), *ATF6* and *BiP* (*HSPA5*) transcript levels were upregulated in CMT2A variant-expressing cells (Fig. 5B). Next, we tested whether ER stress could be the reason for the increase in death of cells expressing CMT2A variants. ER stress triggers several downstream cascades, largely culminating in activation of the ISR pathway. As expected, counteracting the ISR response via addition of the ISR inhibitor ISRIB in WT cells treated with Tg (Sidrauski et al., 2013) reverted the effects of Tg on PARP1 cleavage (Fig. 5A). However, ISRIB did not significantly decrease PARP1 cleavage in MFN2$^{K357N-FLAG}$ cells (Fig. 5C). Similarly, ISRIB did not rescue the ER stress signature for any of the markers observed to be upregulated in MFN2$^{K357N-FLAG}$ cells (Fig. 5B). In conclusion, although MFN2 variants activate ER stress through the unfolded protein response (UPR) pathway, this does not significantly impact their cell death propensity.

## CMT2A variants of MFN2 lead to intrinsic apoptosis

Apoptotic cell death can be caused by both intrinsic and extrinsic factors (Tian et al., 2024). Given that MFN2 locates to mitochondria, and that blocking ER stress responses did not revert PARP1 cleavage in MFN2$^{K357N-FLAG}$ cells (Fig. 5), we decided to test intrinsic apoptosis markers. First, we assessed the localization of the canonical pro-apoptotic BAX protein with immunostaining experiments using an endogenous antibody for BAX, alongside the mitochondrial outer membrane marker FIS1. We could observe a much stronger re-localization of BAX in the MFN2$^{K357N-FLAG}$ cells, consistent with increased intrinsic apoptosis (Fig. 6A; Fig. S5A). We then tested cleavage of the hallmark caspases involved, using western blot analysis. Indeed, caspase 9 cleavage was increased in MFN2$^{K357N-FLAG}$ cells and in MFN2$^{R94Q-FLAG}$ cells (Fig. 6B). Caspase 9 cleavage leads to cleavage of caspase 7 and caspase 3, which were also increased (Fig. 6B). MFN2$^{K357N-FLAG}$ and MFN2$^{R94Q-FLAG}$ cells also presented lower levels of BCL-xL (encoded by *BCL2L1*) (Fig. 6B), consistent with the fact that cleaved and thus active caspase 3 downregulates BCL-xL (Fujita et al., 1998). Nevertheless, cytochrome *c* release could not be detected (Fig. S5B). Interestingly, primary fibroblasts from humans with CMT2A presented increased cleavage of PARP1 and caspase 7 (Fig. 6C), suggesting the physiological importance of apoptosis in a context more proximal to the disease situation. Alongside tubular mitochondrial morphology (Joaquim et al., 2025), no major alterations in cristae structure could be observed in these CMT2A fibroblasts (Fig. S6), once again showing no correlation between cell death and mitochondrial morphology in this context. Taken together, these results allow us to determine that CMT2A variants activate the intrinsic apoptotic pathway.

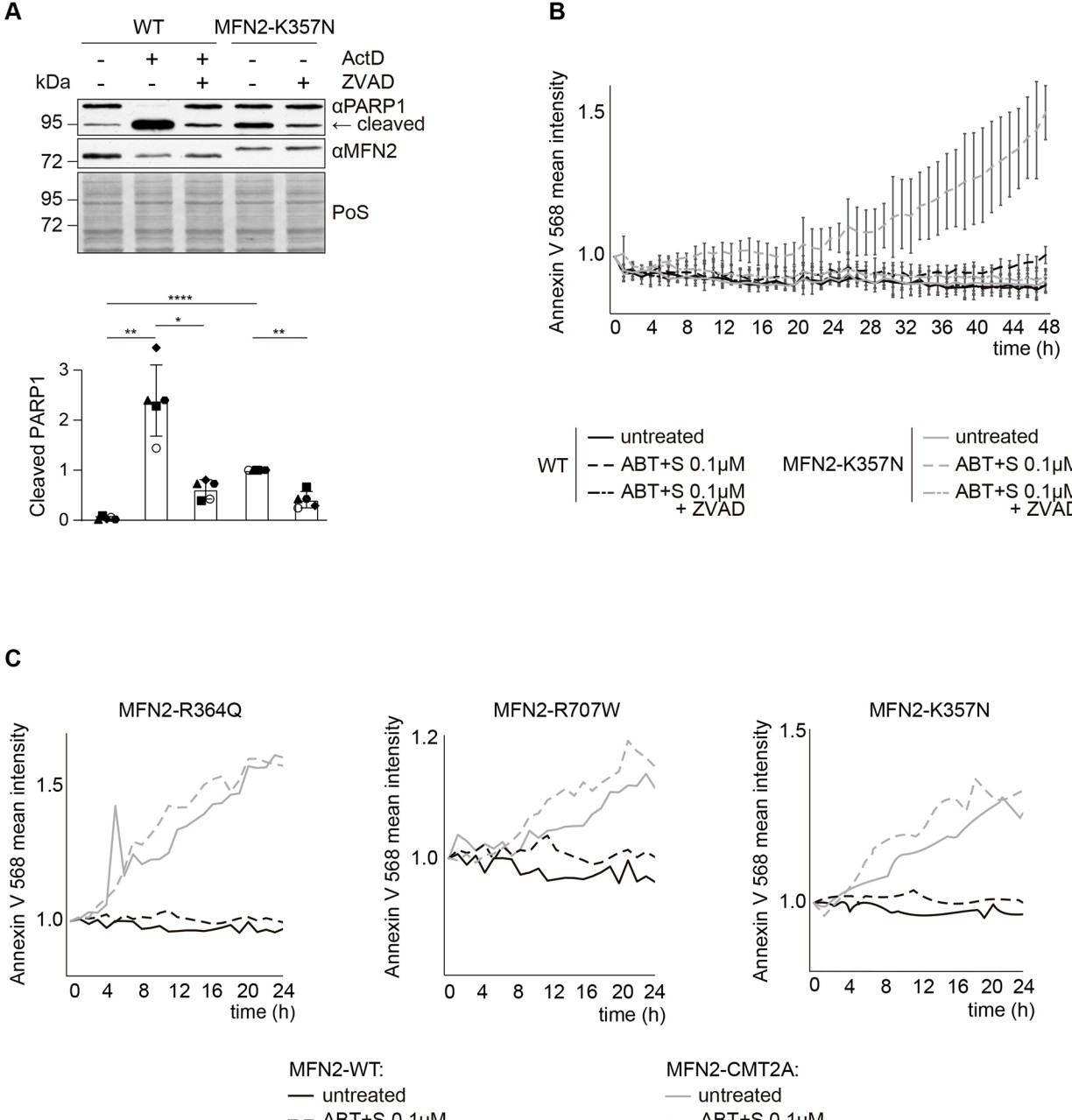

**Fig. 4. CMT2A variants of MFN2 cause apoptotic cell death.** (A) Western blot analysis (upper panel) and quantification (lower panel) of HeLa WT cells or 2KO cells stably expressing FLAG-tagged K357N variant of MFN2, untreated or treated with Act D (1 µM, 6 h), with or without ZVAD (20 µM, 6 h), as indicated, immunoblotted with anti-PARP1 and anti-MFN2 antibodies. Staining of total protein with Ponceau S (PoS) was used as loading control. Bars represent the mean±s.d. intensity of cleaved PARP1, normalized to the PoS staining and shown relative to untreated K357N cells (n=5 biological repeats). Individual values of each experiment are plotted as open and filled circles, squares, triangles, diamonds and hexagons. One-way ANOVA with Tukey's post-hoc test was applied. P values from left to right: **P=0.0091; *P=0.0205; ****P<0.0001; **P=0.0060. (B) Annexin V 568 fluorescence, measured by the Incucyte live imaging system over 48 h and plotted relative to timepoint zero, of HeLa WT or 2KO cells stably expressing FLAG-tagged K357N variant of MFN2, either untreated or treated with ABT+S (BH3 mimetic ABT-737 and the MCL1 inhibitor S63845, 0.1 µM, 48 h) for apoptosis induction, and in presence or absence of ZVAD (20 µM, 48 h) for inhibition of caspases, as indicated. Values represent the mean±s.d. intensity per object count of three technical replicates each from three biological replicates. (C) Annexin V 568 fluorescence, analysed as in B, of HeLa 2KO cells transiently transfected with WT MFN2, MFN2 R364Q, MFN2 R707W or MFN2 K357N, either untreated or treated with ABT+S (0.1 µM, 24 h). Values represent the mean intensity per object count of a single technical replicate each from three biological replicates.

Finally, two reports using mice models have revealed increased accumulation of the pro-inflammatory cytokine interleukin-6 (IL-6) in Mfn2$^{WT/K357T}$ animals compared to WT animals upon treatment with lipopolysaccharide (LPS) (Stavropoulos et al., 2021), alongside suppression of LPS-induced neuroinflammation upon overexpression of WT Mfn2 (Harland et al., 2020). Therefore, we tested whether fibroblasts derived from individuals with CMT2A might exhibit an increase in pro-inflammatory markers. For this, we cultured fibroblasts isolated from healthy donors and individuals with CMT2A harbouring the MFN2 variants R94Q and R94W,

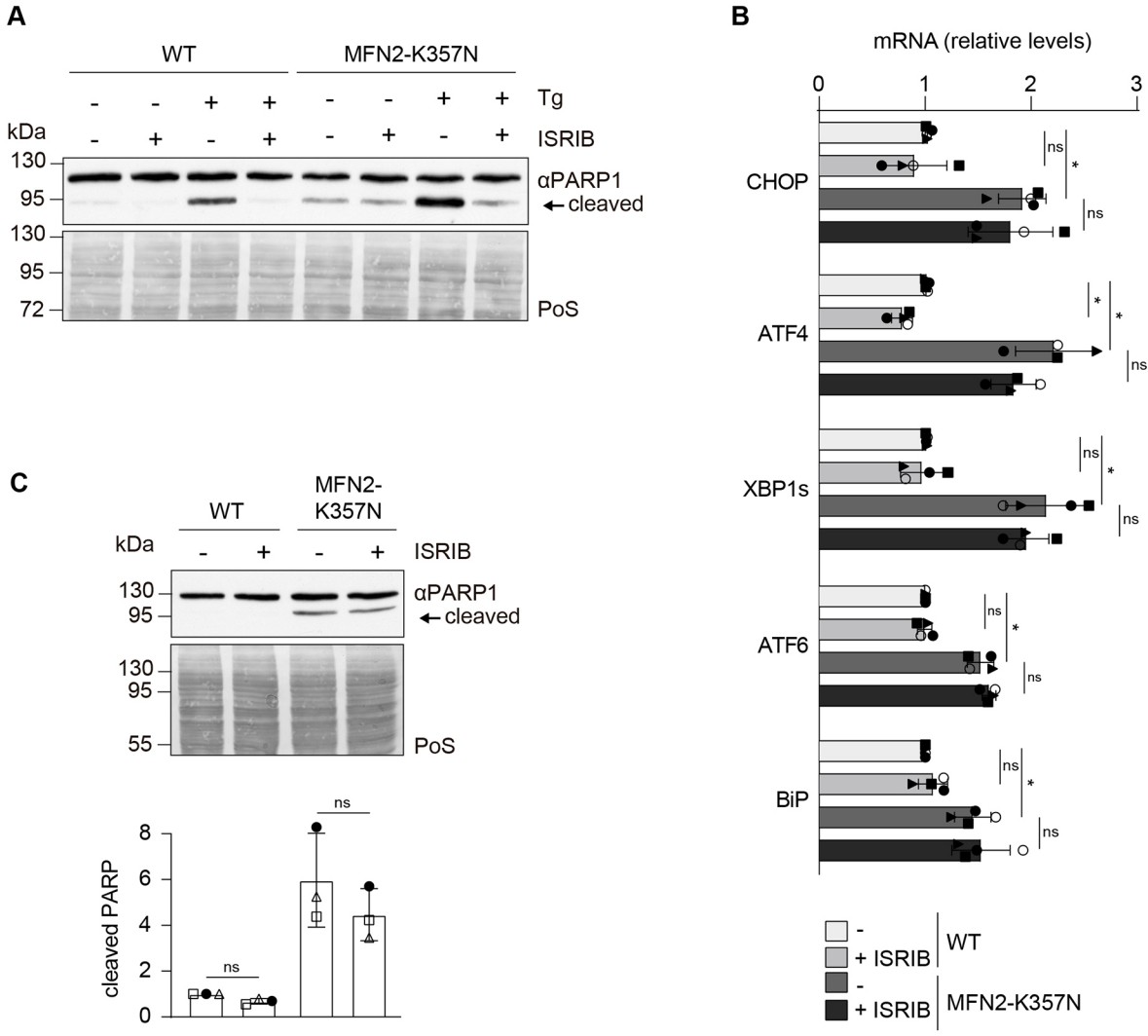

**Fig. 5. The ISR is not implicated in apoptosis observed in cells expressing the K357N variant.** (A) Western blot analysis of HeLa WT cells or 2KO cells stably expressing FLAG-tagged K357N variant of MFN2, untreated or treated with ISRIB (5 μM, 16 h) or thapsigargin (Tg; 0.5 μM, 2 h) as indicated, immunoblotted with anti-PARP1 antibody. Staining of total protein with Ponceau S (PoS) was used as loading control. Blots shown are representative of three experiments. (B) mRNA levels of *CHOP*, spliced *XBP1* (*XBP1s*), *ATF4*, *BiP* and *ATF6* measured by qPCR of total mRNA, extracted from HeLa WT or 2KO cells stably expressing FLAG-tagged K357N variant of MFN2, either untreated or treated with ISRIB (5 μM, 16 h). Transcript levels were quantified using the $2^{-\Delta\Delta Ct}$ method and normalized to *HPRT* transcript levels. Individual values of each experiment are plotted as open or filled circles, squares or triangles. Bars represent the mean±s.d. of the respective mRNA transcript level relative to the untreated WT group (*n*=4 biological replicates). Two-way ANOVA with Tukey's post-hoc test was applied. *P* values for *CHOP* from left to right: ns (not significant), *P*=0.8327; **P*=0.0119; ns, *P*=0.9589. *P* values for *ATF4* from left to right: **P*=0.0488; **P*=0.020; ns, *P*=0.3693. *P* values for *XBP1s* from left to right: ns, *P*=0.9661; **P*=0.0289; ns, *P*=0.8301. *P* values for *ATF6* from left to right: ns, *P*=0.9968; **P*=0.0110; ns, *P*=0.6780. *P* values for BiP from left to right: ns, *P*=0.7441; **P*=0.0420; ns, *P*=0.9608. (C) Western blot analysis (upper panel) and quantification (lower panel) of HeLa WT cells or 2KO cells stably expressing FLAG-tagged K357N variant of MFN2, untreated or treated with ISRIB (5 μM, 16 h), as indicated, immunoblotted with anti-PARP1 antibody. Staining of total protein with PoS was used as loading control. Bars represent the mean±s.d. intensity of cleaved PARP1, normalized to the PoS staining and shown relative to untreated WT (*n*=3 biological replicates). Individual values of each experiment are plotted as open and filled circles, triangles and squares. One-way ANOVA with Tukey's post-hoc test was applied. *P* values from left to right: ns, *P*=0.1082; ns, *P*=0.3795.

collected the conditioned culture medium and measured the level of the cytokine IL-6 as a general marker for tissue inflammation. The results did not support the idea that individuals with CMT2A had generalized increased levels of IL-6 (Fig. S7A). To nevertheless examine inflammation in our HeLa CMT2A models, we performed immunostaining against the p65 subunit of NF-κB (also known as RELA) – a transcription factor central to inflammatory and immune responses. This factor undergoes nuclear translocation when inflammation is induced (Furthmann et al., 2023), as could be observed in response to TNFα treatment (Fig. S7B). Likewise, no

difference in p65 localization was observed between the WT HeLa cells and the cells expressing CMT2A variants. The induction of apoptosis by inflammation, if observed, would be via the extrinsic pathway. Thus, the absence of inflammatory signs in our models is consistent with the observed increased intrinsic apoptosis.

## DISCUSSION
The association of MFN2 with several different pathologies, including neuropathies and metabolic dysfunctions, underlines the utmost importance in understanding disease-relevant functions of

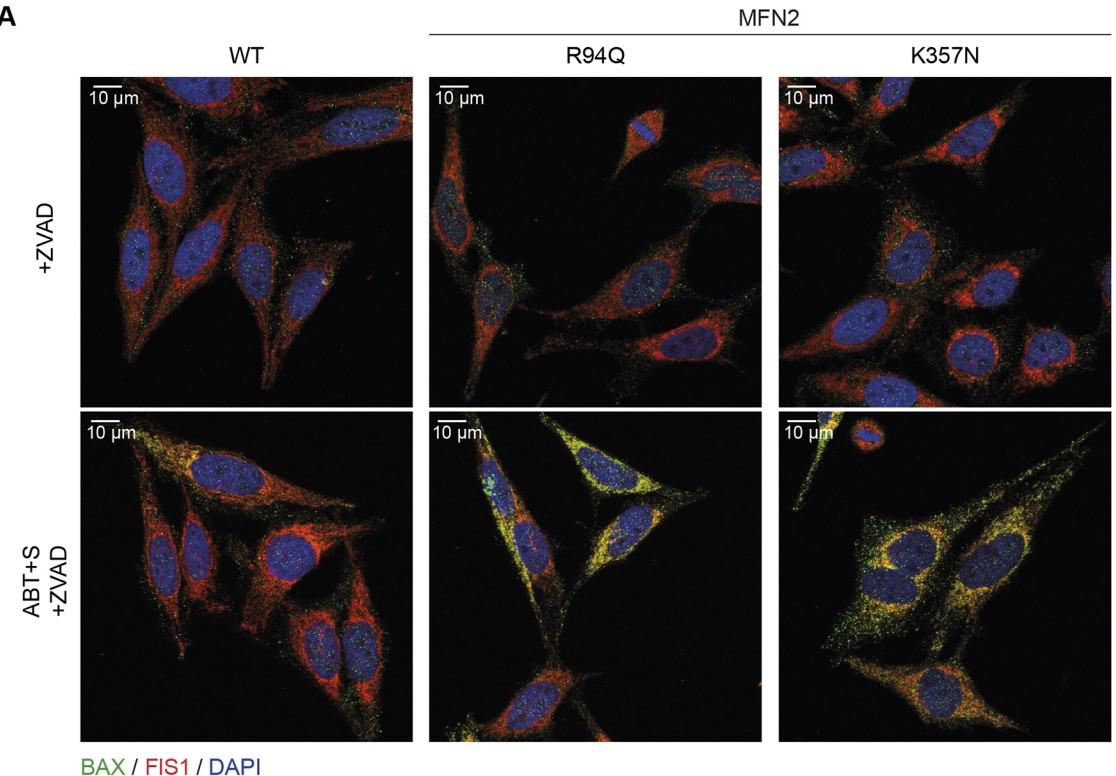

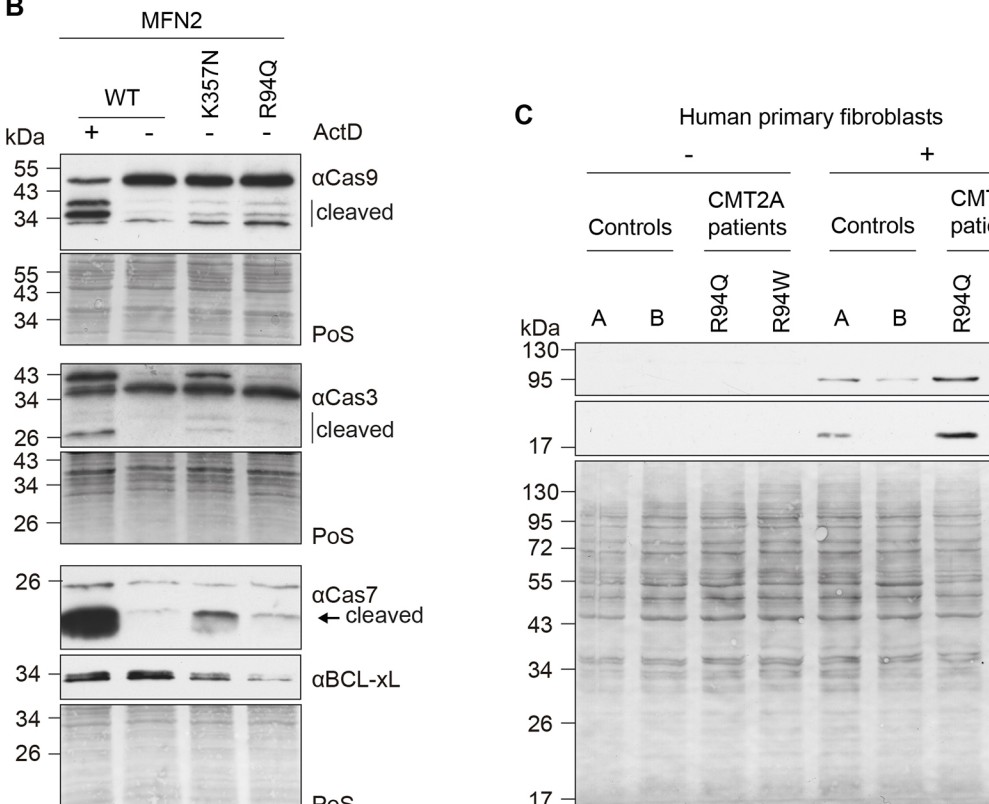

**Fig. 6.** See next page for legend.

**Fig. 6. Caspase cleavage is increased in CMT2A HeLa cells and in fibroblasts from individuals with CMT2A.** (A) Confocal images after immunostaining with monoclonal anti-BAX (in green) antibody, an antibody to detect the outer mitochondrial membrane protein FIS1 (in red) and DAPI (in blue), showing HeLa 2KO cells stably expressing FLAG-tagged WT MFN2 or MFN2 variants (R94Q or K357N), treated with ZVAD (30 μM, 2 h) and either untreated or treated with ABT+S (0.1 μM, 2 h). Scale bars: 10 μm. Images are representative of two experiments. (B) Western blot analysis of 2KO cells stably expressing FLAG-tagged WT, K357N or R94Q variants of MFN2, either untreated or treated with ActD (1 μM, 6 h), as indicated, immunoblotted with anti-caspase 9, anti-caspase 3, anti-caspase 7 and anti-BCL-xL antibodies. Staining of total protein with Ponceau S (PoS) was used as loading control. Blots shown are representative of three experiments. (C) Western blot analysis of two control primary fibroblast lines and two CMT2A primary fibroblast lines carrying the MFN2 variants R94Q or R94W, untreated or treated with staurosporine (STS; 2 μM, 2 h), immunoblotted with anti-PARP1 and anti-caspase 7 antibodies. PoS was used as loading control. It should be noted that in primary fibroblasts we could only detect cleaved PARP1 signal in the presence of cell death inducers. Blots shown are representative of two experiments.

MFN2 (Joaquim et al., 2025; Joaquim and Escobar-Henriques, 2020). It is remarkable that the most severe cases of the neurodegenerative disease CMT2 are caused by MFN2 variants, which can be found spread throughout the whole gene. However, the lack of genotype–phenotype correlation makes it complex to understand the molecular causes that underlie this disease. Here, we established human cellular models that revealed increased apoptotic cell death upon expression of two different MFN2 disease variants, including the highly prevalent R94Q allele. Importantly, this signature was also present in primary fibroblasts from individuals with CMT2A. Increased apoptotic cell death did not fully correlate with alterations in mitochondrial morphology when comparing the R94Q and K357N variants, suggesting a possible pro-apoptotic but morphology-independent role of MFN2 variants in CMT2A.

## CMT2A variants of MFN2 impact differently on mitochondrial morphology

Impaired mitochondrial morphology is the first guess when studying a disease caused by MFN2 variants. However, and in agreement with previous studies, different CMT2A MFN2 variants result in different mitochondrial morphologies, with fragmentation, hyperfusion and normal tubulation being displayed, irrespective of the MFN2 domain affected (Zaman and Shutt, 2022).

Here, the nine variants created were chosen in order to span different disease severity and frequency, or based on bibliographic knowledge (Abati et al., 2022; Kijima et al., 2005). R94Q is an extremely common variant in CMT2A and represents the most studied variant according to databases available on the Inherited Neuropathy Variant Browser (https://neuropathybrowser. zuchnerlab.net/#/), which gathers information on clinical studies (Bernard-Marissal et al., 2019; Detmer and Chan, 2007; Kumar et al., 2023 preprint; Misko et al., 2010; Wolf et al., 2019). K109R is a CMT2A variant known to affect the GTPase activity of MFN2 and is frequently employed as a control for mitochondrial fragmentation (Ando et al., 2017). K357 of MFN2 corresponds to K464 in the yeast mitofusin Fzo1, a very well-studied residue that is essential for mitochondrial fusion (Anton et al., 2013; Simoes et al., 2018). R707W was the first homozygous CMT2A variant to be reported (Sawyer et al., 2015). Finally, R104W, P251A, R280H, R364Q and M376L were chosen out of the most reported CMT2A MFN2 residues (https://neuropathybrowser.zuchnerlab.net/#/).

Among those nine variants, three – K109R, K357N and R104W – led to a fragmented mitochondrial network, whereas

six – R94Q, P251A, R280H, R364Q, M376L and K707W – rescued mitochondrial tubulation of 2KO cells, albeit to different degrees. Notably, overexpression of Mfn1 or Mfn2 agonist molecules in murine models allows recovery of mitochondrial tubulation and reversal of some Mfn2 knockout or CMT2A defects (Motori et al., 2020; Rocha et al., 2018; Zacharioudakis et al., 2022; Zhou et al., 2019). However, CMT2A R94W, H165R, R274W and T362M variants have no impact on the GTPase activity of MFN2 or further steps affecting its membrane tethering capacity, rather suggesting a fusion-independent mechanism (Beręsewicz et al., 2018; Li et al., 2019; Rasheed et al., 2020). Consistently, fibroblasts from individuals carrying one of the variants T105M, I213T, F240I, V273G or L734V (Amiott et al., 2008), R364W, M376V or W740S (Larrea et al., 2019) show WT-like mitochondrial morphology. Additionally, V69F, L76P, R274Q and W740S variants have further been shown to be fusion competent when expressed in double-mitofusin knockout cell culture models (Detmer and Chan, 2007). Moreover, A383V (Samanas et al., 2020) and T206I (Das et al., 2024) variants result in mitochondrial hyperfusion. In contrast, expression of the variants L76P, R94Q, T105M, M376A, S350P or W740S in mouse or cell culture models (Detmer and Chan, 2007; Franco et al., 2022; Misko et al., 2010; Sloat and Hoppins, 2023; Wolf et al., 2019); D414V or D210V in human fibroblasts (Rouzier et al., 2012; Sharma et al., 2021); and T105M, R274W, H361Y or R364W in motor neurons derived from human fibroblasts (Franco et al., 2020; Beręsewicz et al., 2017) does not support mitochondrial fusion. Therefore, our and others' studies conclude that mitochondrial morphology does not correlate with CMT2A.

## Lysine 357 in MFN2

To investigate the mechanistic basis for the involvement of MFN2 in CMT2A, the variant K357N was particularly interesting to study. Three *de novo* missense variants in this residue – to threonine (K357T), glutamic acid (K357E) or asparagine (K357N) – have been reported in individuals with severe early-onset CMT2A (Abati et al., 2022; Kijima et al., 2005). The heterozygous knock-in MFN2$^{K357T}$ mouse model shows aberrant mitochondrial clustering and abnormal morphology, with swelling and disorganized or absent cristae in the sciatic and optic nerves, a phenotype that progressively increases with age (Stavropoulos et al., 2021). Additionally, these mice are more prone to stress-induced neuroinflammation (Stavropoulos et al., 2021). K357 locates in a highly conserved region immediately downstream of the GTPase and before the HR1 domain of MFN2 (Fig. 1A,B) (Honda et al., 2005). Along with the residues R94 and D221, K357 forms a conserved trilateral salt bridge, which connects the HR1 to the GTPase domain at hinge 2, a region regulating dynamic structural rearrangements (Li et al., 2019). Hinge 2 bending allows MFN2 to be in the fusion-promoting transition state and is required for mitofusin activity in yeast (Anton et al., 2019). This agrees with the disruption of mitochondrial tubulation we observed in MFN2$^{K357N-FLAG}$ cells. It also possibly explains the abovementioned murine phenotypes. Interestingly, mitochondrial clustering caused by expression of the K357T variant in SH-SY5Y neuroblastoma cells is partially rescued by expression of human MFN1 (Stavropoulos et al., 2023). However, no data exists yet on whether this rescue of mitochondrial clustering impacts the phenotypic outcome of the disease. The two remaining reported variants in lysine 357 (K357E and K357N) are, to date, only clinically described (Abati et al., 2022; Kijima et al., 2005). Importantly, the individual carrying the variant K357E presented

severe muscle atrophy, whereas the individual carrying the K357N variant displayed decreased myelinated fibres and no Schwann cell proliferation, both indicative of possible cell death (Abati et al., 2022; Kijima et al., 2005).

## Mitochondrial dynamics alterations per se are not sufficient to affect cell death

The role of mitochondrial fusion or fission in apoptosis has been reported in several cell types, model systems and developmental stages. Here, we found that cells lacking the fusion factors MFN1 or MFN2, or lacking the fission factor DRP1, resembled WT cells in regard to PARP cleavage. This means that alterations in mitochondrial morphology and dynamics per se are not sufficient to induce apoptosis in our model. In agreement with our findings, different experimental conditions have revealed that cells harbouring extensively fragmented mitochondrial networks present no alterations in viability (Chen et al., 2003; Karbowski et al., 2006; Lee et al., 2004; Legros et al., 2002; Lim et al., 2001). Nevertheless, mitochondrial fragmentation, the most notable hallmark of depleting mitofusins, or of inducing DRP1, has been correlated with apoptosis induction in several contexts, including embryogenesis, spermatogenesis and cancer (Chen et al., 2020; Choudhary et al., 2014; Frank et al., 2001; Huang et al., 2007; Jenner et al., 2022; Karbowski et al., 2002; Zhao et al., 2015). In a neuronal context, hippocampal-specific depletion of *Mfn2* leads to neuronal cell death in mice, and cultured sensory mouse neurons depleted of Mfn2 show a dying-back degeneration phenotype from distal to axonal regions (Kumar et al., 2023 preprint). Additionally, Mfn2 has been found to be essential for development and maintenance of the cerebellum by promoting Purkinje cell survival, whereas its loss leads to degeneration of these neuronal cells (Chen et al., 2007). Moreover, rat primary neurons with Mfn2 downregulation became more sensitive to cell death induced by excitotoxicity (Martorell-Riera et al., 2014), as did SH-SY5Y neuroblastoma cells with MFN2 knockdown when exposed to rotenone (Yang et al., 2018b). However, and in contrast to these pro-survival roles, MFN2 has been also shown to have pro-apoptotic roles (Brooks et al., 2007; Han et al., 2020; Hoppins et al., 2011; Jin et al., 2011; Karbowski et al., 2002; Montessuit et al., 2010; Neuspiel et al., 2005; Sugioka et al., 2004; Wu et al., 2008; Yang et al., 2018a). Finally, in cardiomyocytes, both anti- and pro-apoptotic effects have been reported upon loss of mitofusins, which depends on the developmental stage and stress conditions (Papanicolaou et al., 2011, 2012). In conclusion, the role of mitochondrial fusion or fission in apoptosis regulation is highly context dependent (Joaquim and Escobar-Henriques, 2020). Absence or decrease in the levels of mitofusins and the subsequent changes in mitochondrial morphology and fitness can either sensitize cells to apoptosis or render them more resistant to it, depending on the cell type and developmental stage.

## CMT2A variants of MFN2 lead to increased apoptotic cell death

In contrast to the effects of loss of fusion factors or fission factors, CMT2A MFN2 variants per se lead to increased PARP cleavage, as shown in Fig. 2A. Notably, the contribution of apoptosis to CMT2A onset or development has been little investigated. In our study, increased intrinsic apoptotic cell death appeared as a common signature in cells expressing different CMT2A variants, as revealed by biochemical, ultrastructural and live-cell analysis. Increased cell death occurred at basal levels but could be further enhanced upon stimulation of apoptosis. First, PARP1 cleavage, a broadly used

apoptotic marker, alongside increased cleavage of caspases 9, 7 and 3, could be observed in cells expressing MFN2 disease variants. Second, rescue of PARP1 cleavage upon caspase inhibition with ZVAD confirmed a link between CMT2A variants and intrinsic apoptotic cell death. Third, primary fibroblasts from individuals with CMT2A also presented increased cleavage of caspases and PARP1. Consistently, decreasing the relative amount of the CMT2A variant by co-expression of WT MFN2 ameliorated PARP1 cleavage. Importantly, these MFN2 variants locate throughout the coding region and present different mitochondrial morphologies, suggesting a dissociation of CMT2A from the role of MFN2 in mitochondrial fusion.

In agreement with our findings, a study using mouse models carrying the Mfn2 variants R364W, G176S or H165R has demonstrated elevated levels of apoptosis (Zhang et al., 2024). In contrast, two studies have reported a different scenario: in one case, fibroblasts from individuals with CMT2A carrying the variants M21V, A166T or R364Q were found to show no alterations in drug-induced apoptosis (Loiseau et al., 2007). In another case, induced pluripotent stem cell-derived motor neurons originating from individuals carrying the CMT2A variant A383V were found to exhibit decreased cell death (Rizzo et al., 2016). However, survival of the CMT2A neurons was inferred from the amount of motor neurons at a given time, which might not necessarily reflect cell survival (Rizzo et al., 2016). Thus, despite the clear signs of apoptosis from the multiple assays performed in our study, future studies in other contexts will be of paramount importance. In particular, studies with cells of neuronal lineages would be highly valuable.

In conclusion, our findings point towards increased apoptotic cell death as a possible contributor to the aetiology of CMT2A, which is consistent with the progressive nature of this neurodegenerative disease. Moreover, the clinical similarity in CMT patients suggests that the relevance of apoptosis might expand beyond CMT2A. Consistent with this idea, increased apoptosis has also been reported in mouse models of CMT type 1A (Sancho et al., 2001). This underlines the general importance of analysing the potential therapeutic benefits of apoptosis inhibitors in CMT2A and other sub-types of this disease.

## MATERIALS AND METHODS
### Cell culture and treatments
The HEK293 (Invitrogen, R75007) and HeLa (ATCC, CCL-2) cells utilized in this investigation were cultured in DMEM-GlutaMAX medium (61965-026, Gibco), containing 4.5 g/l glucose and supplemented with 10% fetal bovine serum (FBS; A52 56701, Gibco), 100 µM non-essential amino acids (11140-035, Gibco), 1 mM sodium pyruvate, and 1% penicillin-streptomycin (15140-122, Gibco). Primary human dermal fibroblasts from two healthy donors and two CMT2A patients were collected after obtaining written informed consent (Ethics Committee from the Angers University Hospital approval: CPP Ouest 6, Angers, France; identification number, CPP1402 HPS2; declaration number, 21.04.27.3982; authorization number, 2021-A00837-34) and in accordance with the principles expressed in the Declaration of Helsinki. Patients were sampled during a neurological consultation by a clinical neurologist at CHU Angers. A confirmed pathogenic variant was required. No participant was remunerated during this study. The witnesses were granted a travel allowance of €30. Sex determination was indicated in the file by the individual's declaration. Gender was not considered in this study since, to date, no clinical data have established a link between the CMT2A phenotype and gender. The gender of the samples was not taken into account. Fibroblasts were cultured in DMEM F12 medium (11320-033, Gibco) containing 1% 10 µg/ml sodium pyruvate, 1% penicillin-streptomycin, 10% de-complemented FBS (A52 56701, Gibco) and 30% complete AmnioMax C100 medium (17001-074, Gibco). To avoid

bias due to prolonged *in vitro* culture, only cells with a passage number of less than 25 were studied.

The HeLa cell lines depleted of MFN1 or MFN2 (HeLa 1KO or 2KO, respectively) as well as the HEK293 cell line depleted of MFN1 (HEK 1KO) were previously described by us (Joaquim et al., 2025).

Stable complementation of HeLa 2KO cells to create the MFN2$^{FLAG}$, MFN2$^{K357N-FLAG}$ and MFN2$^{R94Q-FLAG}$ cell lines was achieved via lentiviral infection. For this, MFN2–3×FLAG, MFN2-K357N–3×FLAG or MFN2-R94Q–3×FLAG constructs (Table S1), respectively, were cloned into a pLVX-puro plasmid (Table S1). HEK293T cells (Abcam, Ab255593) were transfected with 7 µg of the pLVX-puro vectors carrying MFN2 WT or variant versions and Lenti-X packaging single shots (Takara). At 24 h after transfection, fresh medium was added to the cells. Following 24 h, HeLa 2KO cells were treated with the filtered viral particle-containing medium from the transfected HEK293T cells. More specifically, for a well of a six-well plate, 2 ml of the filtered viral medium from the transfected HEK293T cells was mixed with 1 ml DMEM (61965-026, Gibco) and 2.4 µl polybrene (5 mg/ml; TR-1003-G, Sigma-Aldrich). A mock infection was performed where another well of HeLa 2KO cells was treated with 5 ml DMEM and 12 µl polybrene. At 24 h after infection, selection of the cells was initiated by incubation with 1 µg/µl puromycin-containing medium. Selection medium change was performed daily until all cells from the mock infection well were dead. Finally, a monoclonal selection of the three cell lines was performed, and screening of the clones was performed via western blotting for assessment of MFN2 and FLAG protein levels. The selected clones were confirmed by sequencing.

Stable Flp-In T-Rex HEK293 cell lines expressing the FLAG-tagged MFN2 variant K357N or the WT version were established using the previously published HEK293 2KO cell line (Joaquim et al., 2025). To achieve this, $6×10^5$ cells were seeded per well in a six-well plate and co-transfected with the pcDNA5 plasmid encoding MFN2-K357N–3×FLAG or encoding MFN2–3×FLAG and poG44 (Table S1) at a 1:4 ratio, respectively. The transfection procedure followed the protocol mentioned above. At 24 h after transfection, the cells were expanded into three 15 cm dishes and subjected to a 10 day selection process using hygromycin B-containing medium (100 µg/ml). Following the selection period, individual clones were isolated and expanded further. The clones underwent screening through western blot analysis to assess MFN2 and FLAG protein levels. The selected clones were confirmed by sequencing.

The DRP1 knockdown HeLa cell line was created by disruption of the *DRP1* gene by a double nickase CRISPR-Cas9 strategy, targeting exon 1 with paired single guide RNAs (gRNAs). For this, $4×10^5$ cells were plated per well of a six-well plate and transiently transfected with the plasmid pX459 encoding Cas9 nickase targeted against DRP1 by a guide RNA (construct 1, Table S3). At 24 h after transfection, the cells were incubated with puromycin-containing medium for 2 days. The surviving cells were diluted into single cells on 96-well plates and screened by western blotting for knockdown candidates. DRP1 knockout HeLa cell line was generated in and kindly provided by the laboratory of Thomas langer (Max Planck Institute, Cologne, Germany).

Transient transfection was conducted by introducing 1.5 µg of the respective plasmid DNA (Table S1) into cells using GeneJuice transfection reagent (70967-1ml, Millipore) according to the manufacturer instructions in Opti-MEM reduced serum medium (31985-070, Gibco). Microscopic imaging or harvesting of cells to obtain protein lysate was performed 24–48 h after transfection. Tetracycline (1 µg/ml) was added for 18 h, where indicated in the figure legend, in order to induce exogenous expression of MFN1–FLAG.

The following chemicals were used for cell culture treatments: actinomyin D (Sigma-Aldrich, Schnelldorf, Germany), ABT-737 (MedChemExpress, Sollentuna, Sweden), MCL1 inhibitor S63458 (MedChemExpress Sollentuna, Sweden), ZVAD-FMK (Hölzel, Köln, Germany), ISR inhibitor (ISRIB) (Merck, Hamburg, Germany), thapsigargin (NEB, Frankfurt am Main, Germany), staurosporine (Sigma-Aldrich, Schnelldorf, Germany). All chemicals used for cellular treatments were solubilized in DMSO. The treatments were administered by replacing the culture medium with drug-containing medium or adding the drug to the existing medium with the appropriate final concentrations and for the time duration as specified in the figure legends.

## Immunostaining

For immunostaining, cells were seeded with $1.1×10^5$–$1.4×10^5$ cells per well in six-well culture dishes containing coverslips [in the case of HEK293 cells, the coverslips were pre-coated with 0.1 mg/ml poly L-lysine in phosphate-buffered saline (PBS) (Sigma-Aldrich, Schnelldorf, Germany)]. At 24–48 h after seeding and following drug treatments as and when indicated in the figure legends, the cells were fixed by incubation in growth medium solution containing 4% paraformaldehyde for 15 min at 37°C. Subsequently, the fixative solution was removed, and samples were rinsed three times with PBS. After fixation, the cells were permeabilized with 0.15% Triton X-100 solution in PBS for 15 min or with 0.1% saponin in 3% bovine serum albumin (BSA) in PBS in the case of BAX staining, then washed two times for 20 min with PBS. Blocking was done with 2% BSA solution in PBS for 1 h at room temperature (RT). The coverslips were incubated with the desired primary antibodies (Table S2) for 1 h at RT or overnight, followed by two 20 min washes with PBS. Next, the coverslips were incubated with fluorescently labelled secondary antibodies specific to the primary antibodies (Table S2), along with DAPI (62248, Thermo Fisher; 1:1000) for nuclear staining, for 1 h at RT. Finally, the coverslips were washed twice for 20 min with PBS and mounted onto glass slides using ProLong Gold Antifade mountant (P36930, Thermo Fisher). The acquisition of microscopy images was performed using one of the following confocal microscopes: UltraView Vox (Perkin Elmer), Stellaris 5 (Leica) or TCS SP8 (Leica), equipped with 63× objectives, except for the UltraView Vox, where a 60× objective was used.

## PKmito ORANGE live staining

HeLa cells and fibroblasts were seeded in eight-well glass-bottom dishes (ibidi GmbH, Germany) at a density of 15,000 or 11,000 cells per well, respectively. On the next day, the cells were incubated for 30 min at 37°C with PKmito Orange dye (PKMO; Spirochrome, Switzerland) diluted to 1:1000 in culture medium, followed by three washes with medium. Live imaging was performed on the following day on a TCP SP8 gSTED (Leica Microsystems) using PL Apo ×100/1.40 Oil objective. PKMO was excited at 561 nm wavelength and stimulated emission depletion (STED) microscopy was performed using a pulsed depletion laser at 775 nm wavelength.

## TMRE live staining

Cells were stained with tetramethylrhodamine ethyl ester (TMRE; 20 nM) for 30 min at 37°C. TMRE was washed twice with phenol-free DMEM containing all required supplements as detailed above. The cells were then live imaged in a confocal UltraView Vox (Perkin Elmer) microscope with a 60× objective. TMRE staining was used for further automated mitochondrial morphology quantification described below.

## Electron microscopy

For electron microscopy analysis, the cells were cultured on Aclar foil coverslips until ~40% confluency. The cells were fixed using a two-step procedure: 30 min at RT, followed by an additional 30 min at 4°C, with fixation buffer (2% glutaraldehyde, 2.5% sucrose, 3 mM $CaCl_2$ and 100 mM HEPES at pH 7.4). After fixation, the cells were washed three times with 0.1 M sodium cacodylate buffer. Subsequently, they were incubated in 1 M sodium cacodylate buffer containing 1% osmium tetroxide, 1.25% sucrose, and 1% potassium ferricyanide for 1 h at 4°C and next washed three more times with 0.1 M sodium cacodylate buffer. This was followed by a series of incubations in ethanol (50%, 70%, 90% and 100%) for 7 min at 4°C. Two additional incubations were performed using a mixture of EPON resin (E14121, Science Services) and ethanol. The first incubation was performed using a 1:1 ratio of EPON to ethanol for 1 h at 4°C, while the second incubation was made at a 3:1 ratio of EPON to ethanol for 2 h at 4°C. Finally, the cells were embedded in an EPON buffer for 72 h at 62°C. Ultrathin sections with a thickness of 70 nm were obtained using an ultramicrotome (UC6, Leica). and contrasted with 1.5% uranylacetate aqueous solution for 15 min at 37°C. Afterwards, the sections were washed five times with water and incubated for 4 min in lead citrate. Another round of five washes with water was performed before the sections were dried on filter paper. Imaging of the samples was carried out using a transmission

electron microscope (JEM 2100 Plus, JEOL) equipped with a OneView 4 K camera (Gatan) operating at 80 kV and at room temperature, by the CECAD Imaging Facility.

## Western blotting

For western blotting, the cells were harvested from culture plates using a cell scraper using ice-cold PBS and centrifuged at 1000 $g$ for 5 min at 4°C. The cell pellets were washed once again with ice-cold PBS. The pellets were resuspended in an appropriate amount of RIPA lysis buffer [1% Triton X-100, 0.1% SDS, 0.5% sodium deoxycholate, 1 mM EDTA, 50 mM Tris-HCl (pH 7.4), 150 mM NaCl and freshly added Roche cOmplete EDTA-free protease inhibitor]. In the case of fibroblasts, the resuspended cells were subjected to brief sonication (2–3 s) using a hand-held sonicator tip and Bandelin Sonopuls mini20 instrument at 30% amplitude. Subsequently the samples were shaken on a vibrating platform at 1500 r.p.m. for 45 min at 4°C and centrifuged at 16,800 $g$ for 15 min at 4°C. The resulting supernatant was collected. The protein concentration was determined using the Bradford protein assay (Bio-Rad), following the manufacturer's protocol. For subsequent analysis, 150 µg of protein was denatured in 4× Laemmli buffer [1 M Tris-HCl pH 6.8, 60% glycerol, 4% SDS, 0.1% Bromophenol Blue and freshly added 0.1 M dithiothreitol (DTT)] at 1400 r.p.m. in a benchtop shaking platform for 20 min at 45°C. The denatured proteins were separated by SDS-PAGE and transferred onto nitrocellulose membranes. Ponceau S staining was used to assess protein loading. The membranes were blocked in Tris-buffered saline (TBS) supplemented with 5% non-fat milk for 1 h. Subsequently, the membranes were incubated overnight at 4°C with the desired primary antibodies in 5% milk solution (Table S2). After incubation, the membranes were washed three times with TBS and incubated for 1 h with the respective HRP-conjugated secondary antibodies in 5% non-fat milk in TBS. Following three additional washes with TBS, the membranes were developed with chemiluminescent detection using homemade ECL (consisting of 100 mM Tris-HCl pH 8.5, 0.44% luminol, 0.009% $P$-coumaric acid and 0.018% $H_2O_2$), Advansta WesternBright ECL, Amersham ECL or SuperSignal West Atto ECL. The signals on Fujifilm SuperRX were developed using a URIX60 film processor. Uncropped western blot and Ponceau S staining data are shown in Fig. S8.

## RNA isolation and quantitative PCR

Total RNA was extracted from the cells using TRIzol reagent (Thermo Scientific, Dreieich, Germany) following the manufacturer's instructions. For real-time PCR analysis, 500 ng of RNA was subjected to reverse transcription into cDNA using SuperScript III Reverse Transcriptase and oligo(dT) primers from Thermo Scientific. Real-time quantitative PCR was carried out using GoTaq qPCR Master Mix (Promega, Walldorf, Germany), following the manufacturer's recommended protocol. The primer sets used are detailed in Table S3. All experiments were conducted in triplicate on either a CFX96 or CFX384 Thermo Cycler (Bio-Rad, Hercules, Feldkirchen, Germany). Transcript levels were quantified using the 2-ΔΔCt method (Livak and Schmittgen, 2001) and were normalized relative to *HPRT* mRNA transcript levels.

## Incucyte live imaging

To measure cell death rates over a period of time, the cells were incubated and imaged by the Incucyte live imaging system. For this, $1 \times 10^4$ cells/ml were plated in 48-well plates. The next day, apoptosis was induced by a combination of the BH3 mimetic ABT-737 and the MCL-1 inhibitor S63845, at the concentrations and times indicated in the figure legends. For caspase-inhibition, cells were treated with ZVAD-FMK. Additionally, cells were stained with the apoptosis dye Annexin V 568 (1:200 dilution in medium; 4641, Sartorius). Treatments and staining were performed by medium change. Immediately after, the cells were transferred to the Incucyte live imaging system, where they were imaged at every 1 h, for 48 h, with a 20× objective. Image analysis was performed using the Incucyte 2020B software.

## Cytokine analysis of cell culture supernatants

Conditioned fibroblast cell culture supernatant samples from passage 11–12 control or CMT2A samples (R94Q, R94W) were subjected to cytokine analysis (Bio-Rad Laboratories, USA) to measure the concentration of the IL-6 cytokine. Supernatants were thawed and measured in duplicates in a multiplex analyser (Bio-Plex 200, Bio-Rad Laboratories) according to the manufacturer's instructions. The software Bioplex Manager 6.1 (Bio-Rad laboratories) was used to compare the median fluorescence intensity of the samples to the standard curves in order to determine the absolute concentration of the cytokine (pg/ml).

## Quantification and statistical analysis

Quantification of the western blotting results was conducted using Fiji software relative to total protein loading as measured via Ponceau S staining. The quantifications of mRNA or protein levels are presented as mean± s.d. for multiple biological replicates ($n$) as indicated in the figure legends.

Mitochondrial morphology was quantified by eye and categorized as tubular, intermediate or fragmented. At least 50 cells of three biological replicates were counted. Results are presented as mean percentage of cells with each mitochondrial morphology±s.d.

Mitochondrial length and number of mitochondrial junctions of regions of interest (ROIs) of at least ten cells were quantified using the ImageJ/Fiji macro MITOMAPR, following the previously described instructions (Zhang et al., 2019). Mitochondrial length (micrometres/ROI) and number of mitochondrial junctions (counts/ROI) are presented as mean±s.d.

All results were plotted using GraphPad Prism software version 9.0.0, and statistical analysis was done using statistical tests as specified in the figure legends. Statistical analysis is displayed as non-significant (ns) with $P>0.05$, *$P<0.05$, **$P<0.01$, ***$P<0.001$ and ****$P<0.0001$. Exact $P$-values are presented within the figure legends.

## Acknowledgements

We are thankful to Rene Neuhaus, Javier Franco, Marilena Boβ and Ta-Chieh Chen for their help with cloning and analysis of CMT2A variants. We are also thankful to Vincent Anton for MFN2 modelling from the BDLP structure. We would like to thank Professor J. Cassereau, Angers Hospital, Neurology department for the human primary skin fibroblasts. We are grateful to Ana García-Saéz (Institute for Genetics, CECAD, Cologne) for technical advice and generosity, and to the García-Saéz lab for their availability and technical support with the Incucyte Live Imaging System experiments. Additionally, we are thankful to Margarete Odenthal (Institute for Pathology, Medical Faculty and Cologne University Hospital) for usage of the qPCR instrument and critical reading of the manuscript; to Christian Jüngst, Beatrix Martiny and the CECAD Imaging Facility for their excellent assistance; and to Pascal Fischer and Julia Benecke for the cytokine analysis. Finally, we would like to thank Hamid Kashkar (Institute for Molecular Immunology, CECAD, Cologne) for technical advice and the generous gift of antibodies for apoptotic markers and Alessandro Annibaldi (Center for Molecular Medicine Cologne) for Annexin V 568.

## Competing interests

The authors declare no competing or financial interests.

## Author contributions

Conceptualization: M.E.-H., M.J., S.A., A.C.; Data curation: M.J., M.-B.B., M.A.M., L.O., S.A., S.P.; Formal analysis: M.J.; Funding acquisition: M.E.-H., M.J., M.-B.B.; Investigation: M.J., M.-B.B., M.A.M., S.P., L.O., S.A., E.M.; Methodology: M.J., M.-B.B., M.A.M., S.P., L.O., S.A., E.M., A.C., M.E.-H.; Project administration: M.E.-H., M.J., A.C.; Supervision: M.E.-H., M.J., S.A., A.C.; Validation: M.E.-H., M.J., M.-B.B., S.P., S.A., A.C.; Visualization: M.J., M.-B.B., L.O., S.A.; Writing – original draft: M.E.-H., M.J.; Writing – review & editing: M.E.-H., M.J., M.A.M.

## Funding

This work was supported by the Center for Molecular Medicine Cologne (CMMC) (CAP14 and RPA02), by the Fritz Thyssen Foundation (10.15.1.018MN), by the Boehringer Ingelheim Foundation (Plus 3 program), by the Deutsche Forschungsgemeinschaft (DFG; in the frame of SPP2453, project number 541758846; in the frame of CRC 1218 TP A03, to M.E.-H.; a Qualification fellowship to M.J.; and in the frame of Cologne Graduate School of Ageing, to M.-B.B.). Finally, M.J. was also supported by the Otto-Bayer fellowship from the Bayer Foundation. Open Access funding provided by the University of Cologne. Deposited in PMC for immediate release.

## Data and resource availability

All relevant data and details of resources can be found within the article and its supplementary information.

**Peer review history**

The peer review history is available online at https://journals.biologists.com/jcs/lookup/doi/10.1242/jcs.263691.reviewer-comments.pdf

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
