## [Peer Review File · Journal of Cell Science]

Charcot-Marie-Tooth type 2A variants of mitofusin 2 sensitize cells to apoptotic cell death

Mariana Joaquim, Maria-Bianca Bulimaga, Marie A. Mohn, Solenn Plouzenec, Leon Osinski, Selver Altin, Esther Mahabir, Arnaud Chevrollier and Mafalda Escobar-Henriques
DOI: 10.1242/jcs.263691

Editor: Ana Garcia-Saez

Review timeline

Original submission:	4 November 2024
Editorial decision:	16 December 2024
First revision received:	18 June 2025
Editorial decision:	17 July 2025
Second revision received:	18 August 2025
Accepted:	27 August 2025

Original submission

First decision letter

MS ID#: jcs.263691

MS TITLE: Charcot Marie-Tooth Type 2A mutations in Mitofusin 2 sensitize cells to apoptotic cell death

AUTHORS: Mafalda Escobar-Henriques Dias; Mariana Joaquim; Selver Altin; Leon Osinski; Maria-Bianca Bulimaga

ARTICLE TYPE: Research Article

Dear Mafalda,

We have now reached a decision on the above manuscript.

To see the reviewers' reports and a copy of this decision letter, please go to:

As you will see, the reviewers raise a number of substantial criticisms that prevent me from accepting the paper at this stage. They suggest, however, that a revised version might prove acceptable, if you can address their concerns. If you think that you can deal satisfactorily with the criticisms on revision, I would be pleased to see a revised manuscript. We would then return it to the reviewers.

Reviewer 1

Joaquim et al described Charcot Marie-Tooth Type 2A mutations in Mitofusin 2 sensitize cells to 3 apoptotic cell death. I have some questions as following.

1.MFN2 mutation is a fusion protein gene that affects cellular autophagy. Does it affect other fusion proteins OPA1 and MFN1?Does it also affect the levels of related fission proteins? Please provide relevant experimental evidence.

2. MFN2 affects cell apoptosis, detected only by Annexin V. Does it increase immune fluorescence and Western blot assays (apoptosis inhibited protein (Bcl xL) and eight activated proteins (BAD, BAX, uncleared/cleared caspases 3, 7, and 9)) to determine which apoptotic pathway is activated?

3. Can MFN2 mutations activate ER stress through the UPR pathway? Or are there other ways? Please add relevant data.

Reviewer 2

SUMMARY OF THE ADVANCE MADE IN THIS PAPER AND ITS POTENTIAL SIGNIFICANCE TO THE FIELD

As the authors mention, MFN2 has already been implicated in apoptosis. In this small manuscript, they analyze the impact of CMT2A mutations of MFN2 on apoptotic cell death. They claim that CMT2A mutations, including mutations that do not induce mitochondrial fusion defects (based on mitochondrial morphology quantification), induce apoptosis based on PARP1 cleavage, transmission electron microscopy analysis and Annexin V staining. Together, the added scientific value of this work is incremental. Conclusions result from massive overinterpretation of the data, key controls are often missing and the experimental design does not recapitulate the setting of patient cells. Mechanistic data explaining the proposed link between MFN2 mutations and apoptosis is also lacking. The current study is not recommended for publication in Journal of Cell science.

SUGGESTIONS TO AUTHORS

Main comments:

In mitochondrial morphology experiments (Fig. 1B, S1C and 2A), what is the intermediate morphology? Intermediate is not informative and should at least be shown and better described.

The authors assume that intermediate mitochondria do not have any fusion defect (such as in R94Q cells for instance). However, careful analysis of Fig. 2A suggests that R94Q have more Intermediate and fragmented mitochondria than WT. In addition, strictly relying on mitochondrial morphology analysis to conclude about affected or unaffected mitochondrial fusion is prone to overinterpretation. Mitochondrial fusion efficiency should be monitored directly. If the R94Q mutation does not affect fusion, why not analyzing this mutation from Fig. 2C to Fig. 3C? Instead, the biggest part of the study focuses exclusively on the K357N mutation which promotes mitochondrial fragmentation likely because of defective mitochondrial fusion.

When comparing WT and MFN2-WT cells in Fig. 2A, it appears that stable expression of MFN2 in 2KO cells does not restore the morphology seen in WT cells. This suggests that stable expression of MFN2 already affects mitochondrial physiology, likely because of decreased MFN2 expression as seen in Fig. 2B. Adding CMT2A mutations may worsen this effect but may not reflect their effects in patient cells because CMT2A is mainly autosomal dominant with a WT and a mutant allele of MFN2. Transfected and stable expressing cells from this study do not recapitulate this heterozygote state, thereby questioning the relevance of the data presented for the etiology of the CMT2A disease. Along this line, the data in Fig. 2E shows that cleavage of PARP1 decreases when mutated MFN2 is co-expressed with WT MFN2.

The electron microscopy data from Fig. 2D are not sufficient to claim that the K357N mutation induces cell shrinkage, loss of cytoplasmic compartmentalization, membrane blebbing and chromatin condensation. At the minimum, more images and statistics are required but these effects may be caused by the preparation of cells for EM. For instance, cell size does not seem affected in fluorescence microscopy images.

Data from Fig. 3A and 3B should be shown on the same blot because cleavage of PARP1 and inhibition by ZVAD in MFN2-K357N seem weak when compared to WT cells treated with ActD. Similarly, Annexin V staining in K357N cells from Fig. S3 seems very weak. This data should include a control of cell death induction (maybe WT cells treated with ActD) so that comparison can be appreciated.

More generally, analyzing mutations of MFN1 as controls would be informative.

Other comments:

There is a high amount of self-citations, even with one manuscript of the group presented as a preprint that is not published or available online.

The authors mention that MFN2 undergo conformational changes from stretched to bent conformation (line 113) without citing any reference. Is this even established or is this just a hypothesis?

The authors indicate that they selected cells expressing the MFN2 mutant variants to similar levels as WT MFN2 (line 135). Do CMT2A mutations induce decreased or increased expression of endogenous MFN2? If so, these changes should be taken into consideration when it comes to investigating the effects of CMT2A mutations on cell physiology.

Reviewer 3

This review refers to the paper submitted by Joaquim et al. to the JCS in December 2024.

The balance between mitochondrial fusion and fission supports neuronal homeostasis, and mutations in both fusion and fission genes lead to neuropathy. Charcot-Marie-Tooth disease type 2A (CMT2A) is an inherited axonal neuropathy caused mostly by dominant mutations in mitofusin 2 (Mfn2). The authors address an important question in the field: what might be the disease mechanism for the pathogenic alleles of Mfn2 that have some kind of fusion capacity? Indeed, there are few reports that clearly demonstrate that some CMT2A-associated variants of Mfn2 can restore, at least partially, mitochondrial fusion in Mfn1/Mfn2 double knockout cells. 1) The authors confirmed that a significant proportion of CMT2A alleles can support mitochondrial tubulation to some extent. 2) The authors proposed that two mutant with presumably different level of fusion capacity similarly lead to increased susceptibility to apoptosis. 3) The authors concluded that the alteration of fusion capacity is not a major component of CMT2A mechanism. 4) The authors proposed that increased sensitivity to a, somehow Mfn2-dependent (but fusion decorelated), apoptotic cell death is a common hallmark of the pathogenic alleles.

Regarding, active CMT2A alleles, there are 3 main hypotheses. These "active" alleles could affect Mfn2 functions that are not related to fusion (ER-mitochondrial contact sites being a prime possibility). These alleles could somehow tilt the fusion/fission balance towards fusion, as they can lead to increased mitochondrial tubulation (Das et al., 2022 and 2024; Codron et al., 2016). Although these alleles are fusion-competent, they have lower activity than wild-type mitofusin.

Discriminating between these hypotheses required careful quantification of mitochondrial length and connectivity. Unfortunately, it has been standard practice in cell culture models to assess mitochondrial morphology categories (filamentous, fragmented...) at the level of a cell population rather than to measure mitochondrial properties (length, number of branches...) in individual cells. This is partly due to the fact that cell transfection leads to very different levels of expression between cells.

The authors followed this classical qualitative methodology and scored the proportion of cells with either tubular, intermediate or fragmented mitochondria. But, we have now reached a point where we need to go deeper in the analysis if we want to make progress in the field of CMT2A.

In the tubular class, mitochondria can be more or less long. In the intermediate class, the cell will contain more or less fragmented mitochondria and more or less tubular mitochondria. The choice of classification for a given cell is not trivial. These difficulties in interpreting mitochondrial morphology can be illustrated by the image of R94Q Fig. 1. It appears that part of the mitochondrial network is tubular (upper part), whereas the rest (lower part) appears aggregated and fragmented. For K357N it is difficult to determine qualitatively whether the mitochondria are fragmented and aggregated or enlarged or both. This would be informative as large rounded mitochondria are observed when the fission/fusion balance is switched towards fusion (see Sezaki's DRP1 papers). In addition, it appears that transfection conditions varies much between

experiments. Indeed, the rescue of mitochondrial tubulation by wild-type Mfn2 is much weaker in Fig. 1 than in Fig. S1.

For the reasons outlined above, the authors should perform more accurate quantifications with precisely controlled expression levels. Since the authors have FLAG staining as an internal control for expression level, they should theoretically be able to measure and compare mitochondrial length and branch number in individual cells. They should also use systematically stable lines with similar expression levels, as shown in Figure 2 for some of the mutants. The crystal clear signal usually obtained in cell culture allows very high resolution imaging without the need for sophisticated microscopes or image processing. Segmentation of mitochondria should then be quite simple (using automatic thresholding or machine learning algorithms such as Weka, which run very well on Fiji) and simple tools are already available to automatically quantify mitochondrial morphology (such as MitoAnalyser again on Fiji). Quantification at the mitochondrial level also has the great advantage of providing data that is statistically valid for the sample size. It should be noted that none of the results presented by the authors have been tested for statistical significance.

Regarding the authors' claim that there is no correlation between fusion activity and sensitivity to apoptosis, it is very important to at least determine whether CMT2A mutants that form tubules produce mitochondria that are on average identical to wild-type controls (2xKO + Mfn2 WT) or longer or shorter. A mutant that appears in Figure 1 or S1 with tubular mitochondria may indeed be less active than wild-type Mfn2, i.e. defective. This seems to be the case for R94Q in this study. Indeed, the data presented by the author in Figure 1 (transient expression) and even more clearly in Figure 2 (stable expression) suggest that R94Q is less active than wild-type Mfn2, since it drives much less tubulation (and more fragmentation). This is in contrast to the claims made by the authors in the text (lines 133, 160, 177). More careful quantifications might clarify this crucial point. On the other hand, some other mutants might be more active. In either case, the fusion/fission balance would be affected and needs to be taken into account in the disease mechanism. Indeed, increased fission and decreased fusion are linked to apoptosis, but increased fusion could also induce cell death via a variety of cellular stresses, in particular inhibition of mitophagy.

General comments:

- Question relevant to the field
- HeLa Mfn1-Mfn2 double KO human cells are very appropriate to determine intrinsic activity of mutant Mfn2
- HeLa Mfn1-Mfn2 double KO human cells do not allow to evaluate the dominant properties of the mutant Mfn2 (impact on WT Mfn2)
- HeLa cells are far from neuronal cells so impact on cell physiology/death should be taken with caution
- In contrast to the authors' claim, R94Q "rescue" of tubulation seems comparable to that of Mfn2 WT on the presented figures
- The lack of correlation between fusion activity and sensitivity to apoptosis claimed in the paper is then questionable.

Major points:

- Improve control of expression levels (stable line or other means)
- Improve image quality and quantification at mitochondrial level
- Perform statistical analysis of data

Minor points:

Amiott 2008 and Larrea 2019: These papers analyse mitochondria in patient fibroblasts where it is known that Mfn1 rescues Mfn2. These two papers do not show that CMT2A mutants encode functional mitochondria.

First revision

Author response to reviewers' comments**Comments from the Reviewers and Point by Point response:**

Reviewer 1:

Joaquim et al described Charcot Marie-Tooth Type 2A mutations in Mitofusin 2 sensitize cells to 3 apoptotic cell death. I have some questions as following.

We are very thankful to the reviewer for the very helpful suggestions, which allowed us to provide a much better revised version of the manuscript.

1. MFN2 mutation is a fusion protein gene that affects cellular autophagy. Does it affect other fusion proteins OPA1 and MFN1? Does it also affect the levels of related fission proteins? Please provide relevant experimental evidence.

We have analysed how MFN2 mutations affect the levels of the mitochondrial fusion factors MFN1 & OPA1 and of the mitochondrial fission factors DRP1 & MFF, as suggested. The results, obtained with the MFN2 KO cells stably expressing either one of the two CMT2A variants, or the WT MFN2, are shown in Figs. 3B and 3C. These experiments revealed that increased cell death as observed in CMT2A variants does not correlate with alterations in the levels of fusion or fission components, as described in lines 185-192.

2. MFN2 affects cell apoptosis, detected only by Annexin V. Does it increase immune fluorescence and Western blot assays (apoptosis inhibited protein (Bcl xL) and eight activated proteins (BAD, BAX, uncleared/cleared caspases 3, 7, and 9)) to determine which apoptotic pathway is activated?

To determine which apoptotic pathway is activated, we have analysed caspase 9,7 and 3, and BCL-xL by Western blot assays, and BAX localization, by immune fluorescence. We could not obtain any reliable signal for BAD with immune fluorescence, so we instead analysed Cytochrome C release. The results, obtained with the MFN2 KO cells stably expressing either one of the two CMT2A variants, or the WT MFN2, are shown in Figs. 6A and B and 55A and B. These experiments revealed that CMT2A mutations in MFN2 led to enrichment of BAX at the mitochondria and increased cleavage of all three caspases analysed, together with reduced levels of BCL-xL, and as described in lines 256-264. Moreover, we performed a similar analysis in CMT2A patient fibroblasts, which are presented in Fig. 6C and described in lines 264-267, and which rendered similar results. This nicely allowed to pinpoint intrinsic apoptosis as the activated cell death pathway in our CMT2A model cell lines.

3. Can MFN2 mutations activate ER stress through the UPR pathway? Or are there other ways? Please add relevant data.

We could show that MFN2 mutations activate ER stress, and now added ATF6 to our analyses, to cover the three branches of the UPR. Doing so we found that ATF6-, IRE1- and PERK1-dependent UPR branches are activated (Fig. 5B), as described in line 235-239. As this can induce extrinsic apoptosis, via the integrated stress response (ISR), we also tested the effect of the eIF2alpha inhibitor ISRIB. ISRIB renders cells resistant to the effects of eIF2alpha phosphorylation and thereby potently inhibits ISR (Sidrauski et al., 2015) (Fig. 5A). However, we observed that ISRIB did not significantly reduce the stress markers or the cleavage of PARP1 (Figs. 5A and C), as described in lines 242-245. Finally, to investigate if MFN2 mutations induce inflammation, we measured IL-6 in the supernatant of the CMT2A fibroblasts, and the localization of the transcription factor p65 in CMT2A HeLa cells (Figs. 57A and B), as described in lines 276-282 and 282-287, respectively. However, we could not observe any difference from the WT controls. Together with the results from point 2, we conclude that CMT2A MFN2 mutants induce intrinsic apoptosis, without any visible signs of inflammation.

Reviewer 2:

SUMMARY OF THE ADVANCE MADE IN THIS PAPER AND ITS POTENTIAL SIGNIFICANCE TO THE FIELD

As the authors mention, MFN2 has already been implicated in apoptosis. In this small manuscript, they analyze the impact of CMT2A mutations of MFN2 on apoptotic cell death. They claim that CMT2A mutations, including mutations that do not induce mitochondrial fusion defects (based on mitochondrial morphology quantification), induce apoptosis based on PARP1 cleavage, transmission electron microscopy analysis and Annexin V staining. Together, the added scientific value of this work is incremental. Conclusions result from massive overinterpretation of the data, key controls are often missing and the experimental design does not recapitulate the setting of patient cells. Mechanistic data explaining the proposed link between MFN2 mutations and apoptosis is also lacking. The current study is not recommended for publication in Journal of Cell science.

We would like to thank the reviewer for pointing out that our study focuses on the impact of CMT2A mutations of MFN2 on apoptotic cell death, which has been little investigated, when compared to better studied implications of the levels of MFN2 WT in apoptosis. Importantly, we now supply a large number of additional pieces of mechanistic data, which allowed to confirm increased intrinsic apoptosis in CMT2A variants, without any major inflammatory or integrated stress response involvement, and unrelated to mitochondrial morphology.

We have thoroughly addressed all points raised by the reviewer and present an extensively revised version of our manuscript. Our results include the key controls pointed out and now investigated apoptosis in primary fibroblasts from CMT2A patients, which recapitulate the setting of patient cells.

SUGGESTIONS TO AUTHORS

Main comments:

In mitochondrial morphology experiments (Fig. 1B, S1C and 2A), what is the intermediate morphology? Intermediate is not informative and should at least be shown and better described.

The zoom-in area included in Fig. 1D was already pointing out to the main category present in each case. We realized that this was not obvious and have amended it in Fig. 1D, thus presenting illustrative examples of the three categories “tubular, intermediate and fragmented” used for the cardinal quantifications of mitochondrial morphology.

The authors assume that intermediate mitochondria do not have any fusion defect (such as in R94Q cells for instance). However, careful analysis of Fig. 2A suggests that R94Q have more Intermediate and fragmented mitochondria than WT.

Our quantifications in Fig. 1D indeed show that R94Q have more intermediate and fragmented mitochondria than WT. We have revised the text and corrected all mentions to the fusion capacity of the CMT2A mutant variants, which we agree was inaccurate and misleading. Moreover, in order to provide a more substantial quantification we now provide automated measurements of mitochondrial morphology using the MitoMAPR Fiji macro (Fig. S2A), which confirms our previous assessment of mitochondrial morphology, and which is also in agreement with the results of the cardinal quantification and respective statistical analysis (Fig. 1D), as described in lines 141-143. Moreover, to gather information on mitochondrial cristae, we provide new images of our CMT2A cell models using the dye PKmito ORANGE (Figs. S2B and Fig. S6).

In addition, strictly relying on mitochondrial morphology analysis to conclude about affected or unaffected mitochondrial fusion is prone to overinterpretation. Mitochondrial fusion efficiency should be monitored directly. If the R94Q mutation does not affect fusion, why not analyzing this mutation from Fig. 2C to Fig. 3C? Instead, the biggest part of the study focuses exclusively on the K357N mutation which promotes mitochondrial fragmentation likely because of defective mitochondrial fusion.

We totally agree with the reviewer. To accommodate this perspective, we have now toned down the conclusions and exclusively referred to the lack of correlation between mitochondrial morphology and PARP1 cleavage. Given that this nicely confirms several previous studies, we opted

to focus on the novel parts of the manuscript with regard to cell death. For example, we now analysed PARP1 cleavage in MFN1 and DRP1 KO or KD cells (Fig. 3A), which confirms the lack of correlation between mitochondrial morphology and PARP1 cleavage, as described in lines 174-178.

We agree that studying cell death in the R94W and the K357N stable cell lines offers advantages, and now performed the revised experiments testing them in parallel (Figs. 3, 6, S2, S3, S6, S7). We had opted to focus on the K357N because it resembles MFN2 KO in regard to the morphology, however not in regard to cell death. Hence, we believe it is the best scenario to investigate roles of MFN2 that are not linked to mitochondrial morphology defects. In addition, this variant is vastly less studied.

When comparing WT and MFN2-WT cells in Fig. 2A, it appears that stable expression of MFN2 in 2KO cells does not restore the morphology seen in WT cells. This suggests that stable expression of MFN2 already affects mitochondrial physiology, likely because of decreased MFN2 expression as seen in Fig. 2B.

We agree with the reviewer that the stable re-expression of MFN2 does not totally restore mitochondrial morphology, which might alter mitochondrial physiology. However, it is very clear that this does not increase PARP1 cleavage, which is in contrast to the observations with our CMT2A cell models. It is also the reason why we selected stable cell lines from clones that expressed similar levels of MFN2 WT and CMT2A variants (Fig. 1C). However, we now tested two additional stable clones of the K357N variant, where we could see that despite having different levels of MFN2, PARP1 cleavage intensity is not altered (Fig. S3B), as described in lines 178-182.

Therefore, while we do not wish to exclude the existence of differences between WT and the stable expression of MFN2 in 2KO cells, it is very clear that CMT2A MFN2 variants induce PARP1 cleavage and that this does not occur in WT nor 2KO stably expressing WT MFN2 (Fig.2A).

Adding CMT2A mutations may worsen this effect but may not reflect their effects in patient cells because CMT2A is mainly autosomal dominant with a WT and a mutant allele of MFN2. Transfected and stable expressing cells from this study do not recapitulate this heterozygote state, thereby questioning the relevance of the data presented for the etiology of the CMT2A disease. Along this line, the data in Fig. 2E shows that cleavage of PARP1 decreases when mutated MFN2 is co-expressed with WT MFN2.

We agree with the referee that the relative ratio between the WT MFN2 (or MFN1, as we now also show in Fig. 2B) and mutant MFN2 is of high relevance. By overexpressing wild-type MFN1 or MFN2 in the K357N background cell model, we could indeed reduce the levels of PARP1 cleavage, demonstrating the importance of the relative levels (Fig. 2B), as described in lines 158-159. This comment also prompted us to study the apoptotic phenotypes in a heterozygous and more physiological context, by using patient-derived primary fibroblasts. Our data confirmed increased PARP1 and Cas7 cleavage in these primary cells, supporting a phenotype of increased cell death in CMT2A variants in human fibroblasts (Figs. 6C), as described in lines 264-267. Importantly, we do not intend to claim that we prove increased apoptosis in CMT2A patients and are totally aware this would require investigations in more physiological settings. However, small effects in cell culture might translate in important effects *in vivo*, reinforcing the relevance of our findings for future investigations.

The electron microscopy data from Fig. 2D are not sufficient to claim that the K357N mutation induces cell shrinkage, loss of cytoplasmic compartmentalization, membrane blebbing and chromatin condensation. At the minimum, more images and statistics are required but these effects may be caused by the preparation of cells for EM. For instance, cell size does not seem affected in fluorescence microscopy images.

We agree with the reviewer and, to strengthen our claims, we now tested more cell death markers, which allowed us to pinpoint the involvement of intrinsic apoptosis (Figs. 5, 6, and S5), as described in lines 254-263. We would like to underline that the electron microscopic pictures presented are representative ones of what was observed in terms of cell death signature during the analysis, which is in contrast to the focus of our fluorescence microscopy images. These signs of apoptosis were clearly present in the K357N cells, which was in contrast to WT cells, as described in lines

200-203. Importantly, all samples were handled at the same time and by the same researcher, rendering it unlikely to result from preparation issues. Nevertheless, not enough images were acquired to render statistically meaningful quantifications, hence we show them from a qualitative point of view only. For this reason, we toned down our claims regarding the electron microscopy data from Fig. 2D, now presented in S4A.

Data from Fig. 3A and 3B should be shown on the same blot because cleavage of PARP1 and inhibition by ZVAD in MFN2-K357N seem weak when compared to WT cells treated with ActD. Similarly, Annexin V staining in K357N cells from Fig. S3 seems very weak. This data should include a control of cell death induction (maybe WT cells treated with ActD) so that comparison can be appreciated.

We agree with this comment and we now show it. In fact, Fig. 3A and B were originally in the same blot, but we did not have three biological replicates of the WT controls (untreated, with ActD and with both ActD and ZVAD), reason why we separated them into two panels. We now repeated the experiment and show all conditions in a single panel as well as the quantifications and statistical significance (Fig.4A), as described in lines 203-211. Importantly, we do not aim to say that CMT2A mutants have the same levels of PARP1 cleavage when compared to cells treated with ActD, as this apoptosis inducer unleashes a harsh cell death response that results in the death of all cells within some hours, a phenotype that is much stronger than what we observe in K357N cells. Instead, we claim that CMT2A variants constitutively display PARP1 cleavage, meaning without exogenous cell death chemical inducers, contrarily to WT cells. Additionally, it can be observed that, despite the fact that the initial cleavage degree of WT+ActD and of K357N are different (as mentioned by the reviewer), ZVAD inhibits PARP1 cleavage in both cases.

To make this point clear to the reader, we now included on Fig. S4B the Annexin V staining of the positive control - WT cells treated with 1 μ M ABT+S, which confirms the higher propensity for cell death of the K357N cells.

More generally, analyzing mutations of MFN1 as controls would be informative.

Given that CMT or other disease-associated mutations of MFN1 are not known, we instead analysed MFN1 KO, as well as DRP1 KO and KD cell lines, and observed that increased PARP1 cleavage exclusively occurs in CMT2A MFN2 cell lines (Fig. 3A), as described in lines 175-178.

Other comments:

There is a high amount of self-citations, even with one manuscript of the group presented as a preprint that is not published or available online.

The pre-print manuscript is now accepted and properly cited (Joaquim et al., 2025). Six citations out of the 79 references in total in this manuscript are self-citations. Given that our group works on mitofusins, in our view this doesn't appear excessive. We carefully revisited each of these citations but feel that eliminating them compromises the clarity of the manuscript.

The authors mention that MFN2 undergo conformational changes from stretched to bent conformation (line 113) without citing any reference. Is this even established or is this just a hypothesis?

Although the revised version no longer contains this sentence, this reference was present in the discussion (Li et al., 2019), now on lines 356-358, and corresponds to the manuscript:

Y. J. Li *et al.*, Structural insights of human mitofusin-2 into mitochondrial fusion and CMT2A onset. *Nat Commun* 10, 4914 (2019).

The authors indicate that they selected cells expressing the MFN2 mutant variants to similar levels as WT MFN2 (line 135). Do CMT2A mutations induce decreased or increased expression of endogenous MFN2? If so, these changes should be taken into consideration when it comes to investigating the effects of CMT2A mutations on cell physiology.

We agree that the question of how the stability of CMT2A mutant variants affects CMT2A is a very interesting aspect, which could provide valuable therapeutic hints in some cases, in agreement with our previous study (please see Anton et al. (2023)). However, as shown in Fig. 1C, this aspect does not appear to apply in the current study, because both WT, R94Q and K357N variants have the same MFN2 protein levels. Importantly, only the CMT2A mutant variants, but not WT MFN2, sensitize cells to apoptosis (Fig. 6A, B), as described in lines 254-263. Together with the arguments presented above, the cell death propensity observed in our manuscript appears to be unrelated to the MFN2 levels.

Reviewer 3:

This review refers to the paper submitted by Joaquim et al. to the JCS in December 2024.

The balance between mitochondrial fusion and fission supports neuronal homeostasis, and mutations in both fusion and fission genes lead to neuropathy. Charcot-Marie-Tooth disease type 2A (CMT2A) is an inherited axonal neuropathy caused mostly by dominant mutations in mitofusin 2 (Mfn2). The authors address an important question in the field: what might be the disease mechanism for the pathogenic alleles of Mfn2 that have some kind of fusion capacity? Indeed, there are few reports that clearly demonstrate that some CMT2A-associated variants of Mfn2 can restore, at least partially, mitochondrial fusion in Mfn1/Mfn2 double knockout cells. 1) The authors confirmed that a significant proportion of CMT2A alleles can support mitochondrial tubulation to some extent. 2) The authors proposed that two mutant with presumably different level of fusion capacity similarly lead to increased susceptibility to apoptosis. 3) The authors concluded that the alteration of fusion capacity is not a major component of CMT2A mechanism. 4) The authors proposed that increased sensitivity to a, somehow Mfn2-dependent (but fusion decorelated), apoptotic cell death is a common hallmark of the pathogenic alleles.

Regarding, active CMT2A alleles, there are 3 main hypotheses. These "active" alleles could affect Mfn2 functions that are not related to fusion (ER-mitochondrial contact sites being a prime possibility). These alleles could somehow tilt the fusion/fission balance towards fusion, as they can lead to increased mitochondrial tubulation (Das et al., 2022 and 2024; Codron et al., 2016). Although these alleles are fusion-competent, they have lower activity than wild-type mitofusin.

Discriminating between these hypotheses required careful quantification of mitochondrial length and connectivity. Unfortunately, it has been standard practice in cell culture models to assess mitochondrial morphology categories (filamentous, fragmented...) at the level of a cell population rather than to measure mitochondrial properties (length, number of branches...) in individual cells. This is partly due to the fact that cell transfection leads to very different levels of expression between cells.

The authors followed this classical qualitative methodology and scored the proportion of cells with either tubular, intermediate or fragmented mitochondria. But, we have now reached a point where we need to go deeper in the analysis if we want to make progress in the field of CMT2A.

In the tubular class, mitochondria can be more or less long. In the intermediate class, the cell will contain more or less fragmented mitochondria and more or less tubular mitochondria. The choice of classification for a given cell is not trivial. These difficulties in interpreting mitochondrial morphology can be illustrated by the image of R94Q Fig. 1. It appears that part of the mitochondrial network is tubular (upper part), whereas the rest (lower part) appears aggregated and fragmented. For K357N it is difficult to determine qualitatively whether the mitochondria are fragmented and aggregated or enlarged or both. This would be informative as large rounded mitochondria are observed when the fission/fusion balance is switched towards fusion (see Sezaki's DRP1 papers). In addition, it appears that transfection conditions varies much between experiments. Indeed, the rescue of mitochondrial tubulation by wild-type Mfn2 is much weaker in Fig. 1 than in Fig. S1.

For the reasons outlined above, the authors should perform more accurate quantifications with precisely controlled expression levels. Since the authors have FLAG staining as an internal control for expression level, they should theoretically be able to measure and compare mitochondrial length

and branch number in individual cells. They should also use systematically stable lines with similar expression levels, as shown in Figure 2 for some of the mutants. The crystal clear signal usually obtained in cell culture allows very high resolution imaging without the need for sophisticated microscopes or image processing. Segmentation of mitochondria should then be quite simple (using automatic thresholding or machine learning algorithms such as Weka, which run very well on Fiji) and simple tools are already available to automatically quantify mitochondrial morphology (such as MitoAnalyser again on Fiji). Quantification at the mitochondrial level also has the great advantage of providing data that is statistically valid for the sample size. It should be noted that none of the results presented by the authors have been tested for statistical significance.

Regarding the authors' claim that there is no correlation between fusion activity and sensitivity to apoptosis, it is very important to at least determine whether CMT2A mutants that form tubules produce mitochondria that are on average identical to wild-type controls (2xKO + Mfn2 WT) or longer or shorter. A mutant that appears in Figure 1 or S1 with tubular mitochondria may indeed be less active than wild-type Mfn2, i.e. defective. This seems to be the case for R94Q in this study. Indeed, the data presented by the author in Figure 1 (transient expression) and even more clearly in Figure 2 (stable expression) suggest that R94Q is less active than wild-type Mfn2, since it drives much less tubulation (and more fragmentation). This is in contrast to the claims made by the authors in the text (lines 133, 160, 177). More careful quantifications might clarify this crucial point. On the other hand, some other mutants might be more active. In either case, the fusion/fission balance would be affected and needs to be taken into account in the disease mechanism. Indeed, increased fission and decreased fusion are linked to apoptosis, but increased fusion could also induce cell death via a variety of cellular stresses, in particular inhibition of mitophagy.

We would like to thank the reviewer for sharing our enthusiasm on the importance of studying what might be the disease mechanism for the pathogenic alleles of Mfn2. We are also grateful for the very thorough summary of our main findings. We also thank the reviewer for the helpful suggestions.

We totally agree with the reviewer on the different possibilities explaining how MFN2 might affect mitochondrial morphology and also share the experience that the cardinal/classical classification of mitochondrial morphology is not trivial, especially for groups not very experienced in analysing mitochondrial morphology. Indeed, cells do not present a homogeneous morphology, already within one single cell of stable cells, which is even more exacerbated by variations inherent to transient transfection. Thus, as suggested and further detailed below, in order to address the main points raised by the reviewer, we now focus on WT, R94Q and K357N stable cell lines, which present similar levels of MFN2, and provide an automated quantification of mitochondrial length and number of junctions (Fig. S2A), but also test primary fibroblasts of CMT2A human patients (Figs. 6C, S6, S7A). Moreover, we present many new figure panels, which have largely deepened our study. This allowed to pinpoint intrinsic apoptosis as the activated cell death pathway in our CMT2A model cell lines. Additionally, we analysed PARP1 cleavage in MFN1 and DRP1 KO and KD HeLa cell lines, which corroborated our observation that cell death does not trivially correlate with mitochondrial morphology. Finally, we added statistical analysis of the data.

In sum, we now provide a large number of additional pieces of data and clearly show that CMT2A cells (also from patient fibroblasts) have an increased level of basal intrinsic apoptotic cell death.

General comments:

- Question relevant to the field

We would like to thank the reviewer again for these very valuable comments. We have taken them carefully into consideration in revising this manuscript, which is much more complete, thanks to major dataset additions.

- HeLa Mfn1-Mfn2 double KO human cells are very appropriate to determine intrinsic activity of

mutant Mfn2

- HeLa Mfn1-Mfn2 double KO human cells do not allow to evaluate the dominant properties of the mutant Mfn2 (impact on WT Mfn2)

We agree with the reviewer on the advantages and limitations of double MFN1 and MFN2 knockout cells (DKO), which we created and characterized in our recent manuscript (Joaquim et al., 2025). Importantly, the CMT2A mutants were created based on MFN2 KO (2KO) cell lines; in this manuscript we didn't use the DKO cell lines.

- HeLa cells are far from neuronal cells so impact on cell physiology/death should be taken with caution

The reviewer is totally right, and we mention this point in the discussion (lines 418-421). To improve on this, we have also examined patient-derived fibroblasts as a more physiological model. Our data confirmed increased PARP1 and Cas7 cleavage in these primary cells, supporting a phenotype of increased cell death in CMT2A variants in human fibroblasts (Fig. 6C, lines 264-267). Importantly, we do not intend to claim that we prove increased apoptosis in CMT2A patients and are totally aware this would require investigations in more physiological settings. However, small effects in cell culture might translate in important effects *in vivo*, reinforcing the relevance of our findings for future investigations.

- In contrast to the authors' claim, R94Q "rescue" of tubulation seems comparable to that of Mfn2 WT on the presented figures

We have corrected our statements regarding the mitochondrial morphology rescue of the R94Q mutant variant (lines 128-130 and 141-143).

- The lack of correlation between fusion activity and sensitivity to apoptosis claimed in the paper is then questionable.

We must stress that we totally agree with the reviewer on the many challenges of mitochondrial morphology analysis, with limitations from the experimental setting, but also from both the cardinal and from the automated analysis.

Thus, to strengthen this point, we now analysed PARP1 cleavage in HeLa cells lacking the fusion factor MFN1 (MFN1 KO) or the fission factor DRP1 (DRP1 KD and KO cells) (Figs. 3A, S3A, lines 174-178). In both cases, we could not observe increased cell death. This confirms the lack of correlation between mitochondrial morphology and sensitivity to apoptosis.

Major points:

- Improve control of expression levels (stable line or other means)

We totally agree that the level of expression of mitofusins is carefully and dynamically controlled, both in healthy cells and as a response to environmental perturbations. In this study, we also observed that transient transfection of CMT2A plasmids (now shown in Fig. S1) can indeed result in heterogenous mitochondrial morphologies.

For this reason, we have compared WT, R94Q and K357N stable cell lines that express similar levels of MFN2, but where WT cells diverge from the mutants in PARP1 cleavage (Fig. 1C). Additionally, two different clones of K357N expressing MFN2 in different levels showed comparable levels of PARP1 cleavage (Fig. S3B, lines 180-183). Interestingly, this appears to be titrated and ameliorated by exogenous expression of either MFN1 or MFN2 (Fig. 2B, lines 158-159). Therefore, there appears to be an intrinsic propensity of CMT2A variants to induce cell death, which might be masked upon oligomerization with WT mitofusins.

- Improve image quality and quantification at mitochondrial level

In order to address this concern and as suggested by the reviewer we implemented an automated quantification of mitochondrial morphology, now presented in Fig. S2A, using the MitoMAPR Fiji macro. This means of quantification allowed to confirm the analysis of the reviewer. They are also in line with our previous assessment and cardinal quantification of mitochondrial morphology (Fig. 1D). Indeed, the cardinal quantifications in Fig. 1D show that R94Q have more “intermediate” and “fragmented” mitochondria than WT. We have revised the text and corrected all mentions to the fusion capacity of the CMT2A mutant variants, which we agree was inaccurate and misleading. Moreover, to improve image quality and gather information on mitochondrial cristae, we provide new images of our CMT2A cell models using the dye PKmito ORANGE (Figs. S2B and Fig. 6).

- Perform statistical analysis of data

We have now added statistical analysis of the data.

Minor points:

Amiott 2008 and Larrea 2019: These papers analyse mitochondria in patient fibroblasts where it is known that Mfn1 rescues Mfn2. These two papers do not show that CMT2A mutants encode functional mitochondria.

The referee is right that mitofusin 1 was shown to rescue defects observed in CMT2A patient fibroblast mutants. Specifically, two studies from the Baloh lab (Misko et al., 2012, Zhou et al., 2019) showed that, if expressed at higher levels in neurons mimicking CMT2A, MFN1 can alleviate axonal degeneration of CMT2A mouse models, presumably by interfering with mitofusin oligomerization (Detmer and Chan, 2007). Consistently, we now found that overexpression of MFN1, and also of WT MFN2, could reduce PARP1 cleavage (Fig. 2B and lines 158-159).

We believe the referee refers to the wording in the introduction, are thankful for the comment, and replaced “several others were shown to be fusion-competent” by “several others did not affect mitochondrial morphology”. Indeed, Ammiot 2008 analysed five different CMT2A patient fibroblasts, and Larrea 2019 analysed three different CMT2A patient fibroblasts, and both studies find no impairment of mitochondrial morphology, as also stated in the discussion (lines 333-335). These, and many other studies, are consistent with our findings that CMT2A disease variants do not correlate with altered mitochondrial morphology.

References

- ANTON, V., BUNTENBROICH, I., SIMOES, T., JOAQUIM, M., MULLER, L., BUETTNER, R., ODENTHAL, M., HOPPE, T. & ESCOBAR-HENRIQUES, M. 2023. E4 ubiquitin ligase promotes mitofusin turnover and mitochondrial stress response. *Mol Cell*, 83, 2976-2990 e9.
- DETMER, S. A. & CHAN, D. C. 2007. Complementation between mouse Mfn1 and Mfn2 protects mitochondrial fusion defects caused by CMT2A disease mutations. *J Cell Biol*, 176, 405-14.
- JOAQUIM, M., ALTIN, S., BULIMAGA, M. B., SIMOES, T., NOLTE, H., BADER, V., FRANCHINO, C. A., PLOUZENNEC, S., SZCZEPANOWSKA, K., MARCHESAN, E., HOFMANN, K., KRUGER, M., ZIVIANI, E., TRIFUNOVIC, A., CHEVROLLIER, A., WINKLHOFFER, K. F., MOTORI, E., ODENTHAL, M. & ESCOBAR-HENRIQUES, M. 2025. Mitofusin 2 displays fusion-independent roles in proteostasis surveillance. *Nat Commun*, 16, 1501.
- LI, Y. J., CAO, Y. L., FENG, J. X., QI, Y., MENG, S., YANG, J. F., ZHONG, Y. T., KANG, S., CHEN, X., LAN, L., LUO, L., YU, B., CHEN, S., CHAN, D. C., HU, J. & GAO, S. 2019. Structural insights of human mitofusin-2 into mitochondrial fusion and CMT2A onset. *Nat Commun*, 10, 4914.
- MISKO, A. L., SASAKI, Y., TUCK, E., MILBRANDT, J. & BALOH, R. H. 2012. Mitofusin2 mutations disrupt axonal mitochondrial positioning and promote axon degeneration. *J Neurosci*, 32, 4145-55.
- SIDRAUSKI, C., MCGEACHY, A. M., INGOLIA, N. T. & WALTER, P. 2015. The small molecule

ISRIB reverses the effects of eIF2alpha phosphorylation on translation and stress granule assembly. *Elife*, 4.

ZHOU, Y., CARMONA, S., MUHAMMAD, A., BELL, S., LANDEROS, J., VAZQUEZ, M., HO, R., FRANCO, A., LU, B., DORN, G. W., 2ND, WANG, S., LUTZ, C. M. & BALOH, R. H. 2019. Restoring mitofusin balance prevents axonal degeneration in a Charcot-Marie-Tooth type 2A model. *J Clin Invest*, 129, 1756-1771.

Second decision letter

MS ID#: jcs.263691R1

MS TITLE: Charcot Marie-Tooth Type 2A mutations in Mitofusin 2 sensitize cells to apoptotic cell death

AUTHORS: Mafalda Escobar-Henriques Dias; Mariana Joaquim; Maria-Bianca Bulimaga; Marie A. Mohn; Solenn Plouzennec; Leon Osinski; Selver Altin; Esther Mahabir; Arnaud Chevrollier
ARTICLE TYPE: Research Article

Dear Dr Escobar-Henriques Dias,

We have now reached a decision on the above manuscript.

To see the reviewers' reports and a copy of this decision letter, please go to:

As you will see, the reviewers gave favourable reports but raised some critical points that will require amendments to your manuscript. I hope that you will be able to carry these out because I would like to be able to accept your paper, depending on further comments from reviewers.

Reviewer 1

The author answered the questions raised.

Reviewer 2

In this revision the authors aim at validating their hypothesis that MFN2 CMT2A mutations sensitize cells to apoptotic cell death independently of their impact on mitochondrial morphology. Unfortunately, the study still lacks very important analysis to exclude the role of mitochondrial morphology and dynamics and falls short in providing important control experiments to support its claim.

Main comment

If MFN2 CMT2A mutations induce defects independently of their impact on mitochondrial morphology, this implies that the role of MFN2 in mitochondrial fusion is not implicated. Yet, sensitization to apoptotic cell death is an established effect of mitochondrial fusion inhibition. This study avoids quantifying mitochondrial fusion efficiency in CMT2A mutants, which is mandatory to support the main claims of the study.

Additional comments

PARP1 cleavage in untreated WT cells varies significantly among experiments. Sometimes weak or undetectable (2A; 5A; 5C; S3B) and sometimes stronger (3A; 4A). CMT2A mutants must be included in Fig. 3A. Furthermore, the uncleaved form of PARP1 is weaker and even not detected in 2B and 6C.

In Fig. 4B and 6C, cell death or PARP1 cleavage are only detected in the presence of inducers. Yet, mitochondrial morphology and mitochondrial fusion efficiency are never evaluated in the presence of inducers. This is essential because ABT+5 or STS may increase mitochondrial fusion and morphology defects in CMT2A mutant cells.

Cell death and PARP1 cleavage must be evaluated in the presence of inducers for cells used in Fig. 3A as well as for 2KO cells.

Claims about Electron microscopy data have been removed and EM images have been moved in supplemental figures. However, quantification of the CMT2A phenotype shown in S4A is still lacking but remains required to appreciate its extent in the cell population.

Reviewer 3

The authors have improved image analysis (Fig. S2) and provide statistical test.

I have however still some major concerns (see below) :

A) The claim regarding R94Q fusion capacity Line 140-142 is still too strong and can be misleading for the reader : "Stable expression of WT-MFN2 partially rescued mitochondrial tubulation and the R94Q variant also allowed rescue but to lower levels than WT MFN2 (Fig. 1D, S2). In contrast, cells stably expressing the K357N MFN2 mutant presented fragmented mitochondria (Fig.1D)." This suggests that R94Q is still active and not K357N which is not what the data show. One reader could understand that R94Q fall into the well known "fusion-competent" category of CMT2A alleles (Detmer et al, 2007, Barsa et al., 2025) which is clearly not the case based on the data :

- Fig.1D : MFN2-WT partially restores tubulation (i.e. the formation of tubular mitochondria) in 2 KO cells as 30% of the cells contained "tubular mitochondria" instead of 70% for wild type cells and 0% for 2 KO cells. In contrast only 5% of the MFN2-R94Q cells have tubular mitochondria.

- Fig. S2A junctions : MFN2-WT restores the formation of mitochondrial junctions (absolute value 14) that is a readout of mitochondrial fusion. R94Q also drive junction formation (absolute value 5.5) as well as K357N (absolute value 3) but to a much lower extent.

I think it is important that the reader clearly understand the phenotypes and do not consider R94Q as a fusion competent mutant. Joaquim et al. shows that R94Q might have retained some kind of residual fusion activity and that it is higher than the one of K357N. But it is not comparable to the fusion capacity of previously described fusion competent CMT2A alleles V69F, L76P, R250Q, D274Q and W740S. Importantly, MFN2-R94Q has been largely studied in MFN1-MFN2 double knockout mouse embryonic fibroblasts and has always been described as fusion defective. This is in contrast to V69F, L76P, R250Q, D274Q, W740S which form tubules as efficiently as MFN2-WT (Detmer et al. JCB, 2007, confirmed later by Barsa et al., Scientific Reports, 2025). By the way, Joaquim et al. do not contest or discuss these results as they simply wrote line 337 of the discussion that R94Q was shown previously to be fusion defective.

In conclusion, to be more accurate, better reflect the results and protect the reader from misinterpretation, I suggest to write something like:

"Stable expression of WT-MFN2 partially rescues mitochondrial fusion in double knockout cells as shown by restoration of mitochondrial tubulation, increased mitochondrial junctions and increased mitochondrial length (Fig. S2A). Similar analysis revealed that the R94Q and K357N variants were fusion defective, R94Q retaining however a residual activity (Fig. 1D, S2)."

B) Similarly, the statements made line 298-300 and 301-303 are too strong :

- "Here, we established human cellular models which reveal increased apoptotic cell death as a common feature of MFN2 disease variants." Apoptosis is only studied by Joaquim et al in R94Q (+a bit in R94W) and in K357N , meaning two alleles out of hundreds. In addition, the R94 and K357 are very close in MFN2 structure (Li et al., 2019) and are both part of Hinge 2. Hence, I do not think

that we can generalise too much the involvement of apoptosis in CMT2A disease based on these two alleles.

- "Moreover, this did not correlate with alterations in mitochondrial morphology, pointing to a pro-apoptotic but morphology-independent role of MFN2 variants in CMT2A." Although R94Q seems to have retained a higher residual activity than K357N, it is clear from Fig. 1D and Fig. S2 that they both have decreased fusion capacity. Hence, Joaquim et al. might not have provided insights into the toxic mechanism of fusion competent alleles such as V69F, L76P, R250Q, D274Q and W740S. Hence, we can not conclude from Joaquim et al study that increased apoptotic cell death does not correlate with morphology.

To take into account these two points, the author could write something like :

- "Here, we established human cellular models which reveal increased apoptotic cell death in two different MFN2 disease variants including the highly prevalent R94Q allele".

- "Increased apoptotic cell death did not fully correlated with alterations in mitochondrial morphology when comparing R94Q and K357N variants, suggesting a possible pro-apoptotic but morphology-independent role of MFN2 variants in CMT2A."

Minor comments:

Fig. 1D : R94Q and K357N mitochondrial analysis are not shown on the same graph because they were analysed in two different experiments. The MFN2-WT being shown only for R94Q, this strongly weakens the phenotypic comparison. If the authors have this control it should be added to the figure.

Fig. 1D : for clarity the figure the image and histogram text should mention "2KO, MFN2-WT or R94Q or K357N". Otherwise a reader in a hurry could understand that the construct are expressed in a wild type background and not in 2 KO cells. By the way, Fig. S1 is more clear to that respect "2 KO MFN2 / R94Q".

Second revision

Author response to reviewers' comments

Comments from the Reviewers and Point by Point response:

Reviewer 1:

The author answered the questions raised.

We are very thankful to the reviewer.

Reviewer 2:

In this revision the authors aim at validating their hypothesis that MFN2 CMT2A mutations sensitize cells to apoptotic cell death independently of their impact on mitochondrial morphology. Unfortunately, the study still lacks very important analysis to exclude the role of mitochondrial morphology and dynamics and falls short in providing important control experiments to support its claim.

We are thankful to the reviewer and would like to clarify our aims. Our study is dedicated to the effect of CMT2A-MFN2 mutations in apoptosis. Regarding CMT2A and mitochondrial morphology, it was not our intention to make such an affirmative conclusion, and we have now toned down the link between CMT2A and mitochondrial morphology. In fact, the lack of correlation between CMT2A and

mitochondrial morphology was already reported for multiple mutants in many studies. Nevertheless, we do show that impairment of fusion or fission was not sufficient to induce cell death. The revised manuscript includes in figures 2A and 3A the analysis of PARP1 cleavage in MFN2 or MFN1 KO cells, i.e. cells impaired for mitochondrial fusion, and includes in figure 3A PARP1 cleavage in DRP1 KO cells, i.e. cells impaired for mitochondrial fission. Knockout of any of these factors does not alter PARP cleavage, as described in the chapter entitled “Disruption of fusion and fission does not drive death of HeLa cell models”. These results unequivocally demonstrate that, in our model, alterations in mitochondrial morphology and dynamics per se are not sufficient to induce apoptosis. In contrast, CMT2A-MFN2 mutations per se lead to increased PARP cleavage, as shown in Figure 2A. These aspects have been better clarified in the manuscript text, as further described below.

Main comment

If MFN2 CMT2A mutations induce defects independently of their impact on mitochondrial morphology, this implies that the role of MFN2 in mitochondrial fusion is not implicated. Yet, sensitization to apoptotic cell death is an established effect of mitochondrial fusion inhibition.

Indeed, several studies found that inhibition of mitochondrial fusion sensitise cells to apoptosis. In contrast, several other studies report a decrease in apoptosis upon inhibition of mitochondrial fusion. Thus, fusion factors have been linked to both pro- and anti-apoptotic roles. Most importantly, in agreement with our findings, other studies show that mitochondrial fragmentation alone is not sufficient to trigger cell death, because cells harbouring extensively fragmented mitochondrial networks display no alterations in viability (Chen et al., 2003; Karbowski et al., 2006; Lee et al., 2004; Legros et al., 2002; Lim et al., 2001). Therefore, the relationship between mitochondrial fusion/fission alterations and cell death is context-dependent and inhibition of mitochondrial fusion does not necessarily cause cell death. To render this point clearer, we have extended the description of the reported links between mitochondrial morphology and cell death, in a new sub-chapter of the discussion “Mitochondrial dynamics alterations per se are not sufficient to affect cell death”, in lines 376-409.

This study avoids quantifying mitochondrial fusion efficiency in CMT2A mutants, which is mandatory to support the main claims of the study.

Quantifying the impact of CMT2A-MFN2 mutations in fusion rates is not the aim of our study, and in fact we do not intend to claim it. To better describe our results and conclusions, we now implemented several alterations to the manuscript text, as described above and below, also annotated in yellow in the “highlighted” version.

Importantly, the lack of correlation between mitochondrial morphology and CMT2A has been vastly reported, which was already nicely reviewed, and is now described in more detail in the discussion of our manuscript, in lines 326-347. In addition, our controls with MFN1, MFN2 and DRP1 KO cells, i.e., extreme conditions, also demonstrate that, in our system, alterations of fusion or fission are not sufficient to induce apoptosis. This point is now strengthened in the discussion, in lines 376-409. Thus, we strongly believe that providing measurements for the fusion rates of our CMT2A mutants would not alter the conclusions of our manuscript.

Additional comments

PARP1 cleavage in untreated WT cells varies significantly among experiments. Sometimes weak or undetectable (2A; 5A; 5C; S3B) and sometimes stronger (3A; 4A).

This is indeed correct. The variable levels of PARP1 cleavage in the figures mentioned by the reviewer (2A; 5A; 5C; S3B and 3A; 4A) is dependent on the Western blot exposure time, itself dependent on the result that was to be conveyed. For example, Figure 2A and S3B is rather a low exposure so that it can be appreciated that CMT2A-expressing cell lines have higher cleaved PARP1 levels than the wild type cells. Figure 5A and 5C also display a rather low exposure so that the unchanged level of PARP1 cleavage for the K357N cells upon ISRIB treatment can be appreciated. At this level of exposure, the cleaved band for WT cells was not visible. In contrast, in Figure 3A, we show a stronger exposure, to demonstrate the presence - at a very similar level to WT cells - of

PARP1 cleavage for all the stable cell lines compared therein, while still being able to detect the uncleaved PARP in presence of ActD. In Figure 4A we present an exposure reflecting the different levels of PARP1 cleavage throughout the different conditions, i.e., higher cleaved PARP1 for the untreated K357N, even higher in the presence of ActD, and reduced by ZVAD treatment. Importantly, however, in all cases, detectable and non-saturated signals were used for the accompanying quantification graphs.

CMT2A mutants must be included in Fig. 3A. Furthermore, the uncleaved form of PARP1 is weaker and even not detected in 2B and 6C.

The comparison between WT and CMT2A mutants was shown in multiple other previous blots, therefore we do not see it as necessary to add CMT2A mutants to Figure 3A.

Indeed, the uncleaved form of PARP1 is weaker in figure 2B. This is because transient transfection per se increases the basal stress level and therefore PARP1 cleavage, as can be appreciated in the figure below. In figure 6C, the absence of uncleaved PARP1 is due to the fact that in primary fibroblast cells, which are not comparable to HeLa, we are only able to detect cleaved PARP1 signal, and only in the presence of inducers. This is now clarified in the manuscript, in the respective figure legends, in lines 1023-1024 and 1125-1127, respectively.

Figure: Transient transfection creates cellular stress. Western blot analysis of HeLa WT untransfected (-) or transfected with an empty vector (+), immunoblotted with anti-PARP1. Staining of total protein with PoS was used as loading control.

In Fig. 4B and 6C, cell death or PARP1 cleavage are only detected in the presence of inducers. Yet, mitochondrial morphology and mitochondrial fusion efficiency are never evaluated in the presence of inducers. This is essential because ABT+S or STS may increase mitochondrial fusion and morphology defects in CMT2A mutant cells.

Both figure 4B and 6C compare cell death of CMT2A mutants to the respective WT cells processed under the same treatment conditions, i.e., either by Incucyte cell death measurements in HeLa cells or by PARP cleavage in primary fibroblasts. Therefore, the usage of different assays, performed in different cell lines throughout the whole manuscript, allow us to conclude that CMT2A and apoptotic cell death present a positive correlation. The presence of cell death inducers is indeed associated to an increase in mitochondrial fragmentation, however, for the reasons explained above, mitochondrial fragmentation per se is not sufficient to trigger cell death and, importantly, mitochondrial morphology was not the focus of our manuscript.

Cell death and PARP1 cleavage must be evaluated in the presence of inducers for cells used in Fig. 3A as well as for 2KO cells.

The strength of our manuscript is that, even in the absence of cell death inducers, CMT2A-MFN2 mutations per se lead to increased PARP cleavage, which is not the case of cells lacking MFN2, as shown in Figure 2A. Figure 3A aims at showing that absence of MFN1 or DRP1 is also not sufficient to alter PARP cleavage, resembling MFN2 KO cells. Therefore, we did not see it as necessary to add cell death inducers to either Fig. 2A (MFN2 KO) or 3A (MFN1 and DRP1 KO).

Claims about Electron microscopy data have been removed and EM images have been moved in supplemental figures. However, quantification of the CMT2A phenotype shown in S4A is still lacking

but remains required to appreciate its extent in the cell population.

We apologise that we are not in a position to repeat the EM imaging that is shown in Figure S4A in order to be able to quantify the results. We are convinced that the images that were obtained are an essential piece of evidence showing apoptotic tendencies in untreated CMT2A cells and not in WT cells. Although the data are not sufficient to be quantitative, we received input from cell death colleague experts who assured us that the irregular cell shape seen in CMT2A cells indicate signs of apoptosis.

Reviewer 3:

The authors have improved image analysis (Fig. S2) and provide statistical test.

We are very thankful for the reviewer's acknowledgment of our manuscript's improvement and for very nice and constructive suggestions. Please find below our responses to the points mentioned by the reviewer.

I have however still some major concerns (see below) :

A) The claim regarding R94Q fusion capacity Line 140-142 is still too strong and can be misleading for the reader : "Stable expression of WT-MFN2 partially rescued mitochondrial tubulation and the R94Q variant also allowed rescue but to lower levels than WT MFN2 (Fig. 1D, S2). In contrast, cells stably expressing the K357N MFN2 mutant presented fragmented mitochondria (Fig.1D)." This suggests that R94Q is still active and not K357N which is not what the data show. One reader could understand that R94Q fall into the well known "fusion-competent" category of CMT2A alleles (Detmer et al, 2007, Barsa et al., 2025) which is clearly not the case based on the data :

- Fig.1D : MFN2-WT partially restores tubulation (i.e. the formation of tubular mitochondria) in 2 KO cells as 30% of the cells contained "tubular mitochondria" instead of 70% for wild type cells and 0% for 2 KO cells. In contrast only 5% of the MFN2-R94Q cells have tubular mitochondria.

- Fig. S2A junctions : MFN2-WT restores the formation of mitochondrial junctions (absolute value 14) that is a readout of mitochondrial fusion. R94Q also drive junction formation (absolute value 5.5) as well as K357N (absolute value 3) but to a much lower extent.

I think it is important that the reader clearly understand the phenotypes and do not consider R94Q as a fusion competent mutant. Joaquim et al. shows that R94Q might have retained some kind of residual fusion activity and that it is higher than the one of K357N. But it is not comparable to the fusion capacity of previously described fusion competent CMT2A alleles V69F, L76P, R250Q, D274Q and W740S. Importantly, MFN2-R94Q has been largely studied in MFN1-MFN2 double knockout mouse embryonic fibroblasts and has always been described as fusion defective. This is in contrast to V69F, L76P, R250Q, D274Q, W740S which form tubules as efficiently as MFN2-WT (Detmer et al. JCB, 2007, confirmed later by Barsa et al., Scientific Reports, 2025). By the way, Joaquim et al. do not contest or discuss these results as they simply wrote line 337 of the discussion that R94Q was shown previously to be fusion defective.

In conclusion, to be more accurate, better reflect the results and protect the reader from misinterpretation, I suggest to write something like:

"Stable expression of WT-MFN2 partially rescues mitochondrial fusion in double knockout cells as shown by restoration of mitochondrial tubulation, increased mitochondrial junctions and increased mitochondrial length (Fig. S2A). Similar analysis revealed that the R94Q and K357N variants were fusion defective, R94Q retaining however a residual activity (Fig. 1D, S2)."

Thank you very much for these thorough remarks. We agree with the reviewer and have modified the text. It now reads (lines 140-144):

"Stable re-expression of WT MFN2 partially rescued mitochondrial tubulation (Fig. 1D), and resembled WT cells in regards to mitochondrial junctions and mitochondrial length (Fig. S2A).

Similar analysis revealed that the R94Q and K357N variants had defective mitochondrial morphology, K357N being however more similar to 2KO cells (Fig. 1D, S2)."

B) Similarly, the statements made line 298-300 and 301-303 are too strong :

- "Here, we established human cellular models which reveal increased apoptotic cell death as a common feature of MFN2 disease variants." Apoptosis is only studied by Joaquim et al in R94Q (+a bit in R94W) and in K357N, meaning two alleles out of hundreds. In addition, the R94 and K357 are very close in MFN2 structure (Li et al., 2019) and are both part of Hinge 2. Hence, I do not think that we can generalise too much the involvement of apoptosis in CMT2A disease based on these two alleles.

- "Moreover, this did not correlate with alterations in mitochondrial morphology, pointing to a pro-apoptotic but morphology-independent role of MFN2 variants in CMT2A." Although R94Q seems to have retained a higher residual activity than K357N, it is clear from Fig. 1D and Fig. S2 that they both have decreased fusion capacity. Hence, Joaquim et al. might not have provided insights into the toxic mechanism of fusion competent alleles such as V69F, L76P, R250Q, D274Q and W740S. Hence, we can not conclude from Joaquim et al study that increased apoptotic cell death does not correlate with morphology.

To take into account these two points, the author could write something like :

- "Here, we established human cellular models which reveal increased apoptotic cell death in two different MFN2 disease variants including the highly prevalent R94Q allele".

- "Increased apoptotic cell death did not fully correlated with alterations in mitochondrial morphology when comparing R94Q and K357N variants, suggesting a possible pro-apoptotic but morphology- independent role of MFN2 variants in CMT2A."

Thank you very much for these thorough and nice suggestions, which we entirely implemented as proposed by the reviewer, and can be found in lines 299-301 and 302-305.

Minor comments:

Fig. 1D : R94Q and K357N mitochondrial analysis are not shown on the same graph because they were analysed in two different experiments. The MFN2-WT being shown only for R94Q, this strongly weakens the phenotypic comparison. If the authors have this control it should be added to the figure.

Unfortunately, we do not have this control in this experiment.

Fig. 1D : for clarity the figure the image and histogram text should mention "2KO, MFN2-WT or R94Q or K357N". Otherwise a reader in a hurry could understand that the construct are expressed in a wild type background and not in 2 KO cells. By the way, Fig. S1 is more clear to that respect "2 KO MFN2 / R94Q".

We have followed this suggestion and updated Figure 1D accordingly.

Chen, H., Detmer, S. A., Ewald, A. J., Griffin, E. E., Fraser, S. E., Chan, D. C. (2003). Mitofusins Mfn1 and Mfn2 coordinately regulate mitochondrial fusion and are essential for embryonic development. *J Cell Biol*, 160(2), 189-200.

Karbowski, M., Norris, K. L., Cleland, M. M., Jeong, S. Y., Youle, R. J. (2006). Role of Bax and Bak in mitochondrial morphogenesis. *Nature*, 443(7112), 658-662.

Lee, Y. J., Jeong, S. Y., Karbowski, M., Smith, C. L., Youle, R. J. (2004). Roles of the mammalian mitochondrial fission and fusion mediators Fis1, Drp1, and Opa1 in apoptosis. *Mol Biol Cell*, 15(11), 5001-5011.

Legros, F., Lombes, A., Frachon, P., Rojo, M. (2002). Mitochondrial fusion in human cells is efficient, requires the inner membrane potential, and is mediated by mitofusins. *Mol Biol Cell*, 13(12), 4343- 4354.

Lim, M. L., Minamikawa, T., Nagley, P. (2001). The protonophore CCCP induces mitochondrial permeability transition without cytochrome c release in human osteosarcoma cells. *FEBS Lett*, 503(1), 69-74.

Third decision letter

MS ID#: jcs.263691R2

MS Title: Charcot Marie-Tooth Type 2A mutations in Mitofusin 2 sensitize cells to apoptotic cell death

Authors: Mafalda Escobar-Henriques Dias; Mariana Joaquim; Maria-Bianca Bulimaga; Marie A. Mohn; Solenn Plouzenec; Leon Osinski; Selver Altin; Esther Mahabir; Arnaud Chevrollier

Article Type: Research Article

Dear Mafalda,

I am happy to tell you that your manuscript has been accepted for publication in Journal of Cell Science, pending standard publication integrity checks. It was accepted on 27 Aug 2025. Where referee reports on this version are available, they are appended below.